# Inferred expression regulator activities suggest genes mediating cardiometabolic genetic signals

Jason W. Hoskins[1]*, Charles C. Chung[1,2], Aidan O'Brien[1], Jun Zhong[1], Katelyn Connelly[1], Irene Collins[1], Jianxin Shi[3], Laufey T. Amundadottir[1]*

1 Laboratory of Translational Genomics, Division of Cancer Epidemiology and Genetics, National Cancer Institute, National Institutes of Health, Bethesda, Maryland, United States of America, 2 Cancer Genome Research Laboratory, Division of Cancer Epidemiology and Genetics, National Cancer Institute, National Institutes of Health, Bethesda, Maryland, United States of America, 3 Biostatistics Branch, Division of Cancer Epidemiology and Genetics, National Cancer Institute, National Institutes of Health, Bethesda, Maryland, United States of America

* jason.hoskins@nih.gov (JWH); amundadottirl@mail.nih.gov (LTA)

**Data Availability Statement:** All analysis results are available in the Supporting Information files. No new programs or packages were created for this study, but all scripts used to run the analyses are

## Abstract

Expression QTL (eQTL) analyses have suggested many genes mediating genome-wide association study (GWAS) signals but most GWAS signals still lack compelling explanatory genes. We have leveraged an adipose-specific gene regulatory network to infer expression regulator activities and phenotypic master regulators (MRs), which were used to detect activity QTLs (aQTLs) at cardiometabolic trait GWAS loci. Regulator activities were inferred with the VIPER algorithm that integrates enrichment of expected expression changes among a regulator's target genes with confidence in their regulator-target network interactions and target overlap between different regulators (i.e., pleiotropy). Phenotypic MRs were identified as those regulators whose activities were most important in predicting their respective phenotypes using random forest modeling. While eQTLs were typically more significant than aQTLs in *cis*, the opposite was true among candidate MRs in *trans*. Several GWAS loci colocalized with MR *trans*-eQTLs/aQTLs in the absence of colocalized *cis*-QTLs. Intriguingly, at the 1p36.1 BMI GWAS locus the *EPHB2 cis*-aQTL was stronger than its *cis*-eQTL and colocalized with the GWAS signal and 35 BMI MR *trans*-aQTLs, suggesting the GWAS signal may be mediated by effects on *EPHB2* activity and its downstream effects on a network of BMI MRs. These MR and aQTL analyses represent systems genetic methods that may be broadly applied to supplement standard eQTL analyses for suggesting molecular effects mediating GWAS signals.

## Author summary

The following paragraph is the author summary.

Most human genetic variants lie outside of genes (the functional units of the genome that directly affect a cell's biology) making it unclear which genes are responsible for influencing their associated traits. The gold-standard for linking genetic variants to genes is expression QTL (or eQTL) analysis, which tests for associations between genetic variants

available on GitHub (https://github.com/hoskinsjw/aQTL2021). All data used in the analyses are available from the following sources: • TwinsUK adipose RNA-seq data: The European Genome-phenome Archive EGAS00001000805 (https://www.ebi.ac.uk/ega/studies/EGAS00001000805) • TwinsUK genotype data: Available upon application to the TwinsUK cohort (https://twinsuk.ac.uk/resources-for-researchers/access-our-data/). • TwinsUK phenotype data: Available upon application to the TwinsUK cohort (https://twinsuk.ac.uk/resources-for-researchers/access-our-data/). • METSIM adipose array data: Gene Expression Omnibus GSE70353 (https://www.ncbi.nlm.nih.gov/geo/query/acc.cgi?acc=GSE70353) • GWAS summary statistics are available from GIANT (https://portals.broadinstitute.org/collaboration/giant/index.php/GIANT_consortium_data_files), DIAGRAM (https://diagram-consortium.org/downloads.html), and Global Lipids Genetics (http://csg.sph.umich.edu/willer/public/lipids2013/) consortia.

**Funding:** This work was supported by the Intramural Research Program (IRP) of the Division of Cancer Epidemiology and Genetics (DCEG), National Cancer Institute (NCI), US National Institutes of Health (NIH). The funders had no role in study design, data collection and analysis, decision to publish, or preparation of the manuscript.

**Competing interests:** The authors have declared that no competing interests exist.

and the expression of genes. However, this approach often fails to identify gene(s) potentially mediating the effects of trait-associated variants. Here we propose the use of a supplementary approach called activity QTL (or aQTL) analysis using existing eQTL data. We first inferred the activities of genes that affect other genes' expression based on a gene regulatory network and then tested associations between genetic variants and these inferred regulator activities. This can be advantageous when a gene's measured expression level is a poor indicator of its downstream activity or when multiple genetic influences are funneled through key regulators in a gene regulatory network to affect the trait of interest. Using this approach, we identified genes expressed in adipose tissue (i.e., fat) potentially mediating genetic effects on BMI, fat distribution, diabetes risk and blood cholesterol levels. More broadly, this work highlights the benefits of leveraging relational (i.e., topological) information in addressing complex biological problems.

## Introduction

Genome wide association studies (GWAS) have identified hundreds of thousands of germline variants across the human genome that are associated with thousands of complex traits [1]. However, the complex, ancestry-dependent linkage disequilibrium (LD) landscape and the propensity for genome-wide significant variants to lie in non-coding genomic regions make it challenging to determine functional variants and their target genes. Therefore, the development of empirically supported models explaining how these associations are mediated has lagged further and further behind the discovery of new GWAS signals [2]. An essential element of such explanatory models is the identification of the target gene(s) that are allele-specifically affected by the functional variant(s) underlying a given GWAS signal. In parallel, expression quantitative trait loci (eQTL) studies, wherein germline variants are tested for association with transcript levels in particular tissues or cell types, have discovered millions of variants associated with expression levels of one or more *cis* genes in at least one tissue type [3,4]. The hope has been that comparing GWAS signals to such eQTL results might readily suggest the genes mediating GWAS associations. However, the relevant target genes of GWAS signals have not become as readily apparent from eQTL results as once hoped [4–6].

Simple overlaps between eQTL and GWAS signals are insufficient to implicate target genes mediating the GWAS signal. Colocalization analyses calculate the likelihood of a shared functional variant underlying both the eQTL and GWAS signals, which must be the case if the expression of a gene is mediating any component of the GWAS signal's effect [7]. The Genotype-Tissue Expression (GTEx) Consortium reported a median of ~50% of GWAS loci across 21 traits colocalized with at least one *cis*-eQTL from any tissue in their v6p data freeze [5]. In contrast, in the most recent GTEx data freeze (v8) they reported a median of only 21% of GWAS loci colocalized with at least one *cis*-eQTL from any tissue across 87 traits despite increasing the total samples by over 10,000 (~146%), suggesting the problem of identifying mediating genes through eQTLs may be more subtle than merely a lack of statistical power [4,5]. The potential limitations of *cis*-eQTLs in elucidating GWAS signals was further emphasized by new evidence suggesting on average only 11% of heritability across 42 tested GWAS traits is explained by the *cis* genetic component of gene expression levels [6].

Given the limited success of *cis*-eQTLs in explaining the majority of GWAS signals, methods for determining genetic effects on molecular QTLs further downstream of transcript levels may prove useful. Indeed, enrichment of GWAS variants among significant *trans*-eQTLs tends to be stronger than that of *cis*-eQTLs [3,4]. However, *trans*-QTL analyses suffer from

large multiple testing burdens that limit their power. Reducing tested targets based on regulatory interactions has previously helped identify *trans*-eQTLs [8,9]. While *trans*-eQTLs are enriched for *cis*-eQTLs, only 31.6% of lead *trans*-eVariants are also *cis*-eVariants, though mediation analysis does suggest most of those *trans* effects are likely mediated by the *cis*-eQTL [4]. However, this means the remaining 68.4% of all lead *trans*-eVariants do not coincide with any detected *cis*-eQTL. These unexplained *trans* effects could arise by undetected *cis* effects, direct *trans* effects or more complex, indirect mechanisms.

Among all genetic variants, those with some effect on a complex trait of interest likely have convergent downstream molecular effects that are functionally canalized by cellular regulatory systems [10]. Consequently, we expect functional GWAS variants to frequently have detectable impacts on the activity of regulatory networks in cell types and tissues relevant to the trait of interest, even when more proximal *cis* effects continue to resist identification. Toward this end, we propose the application of activity QTL (aQTL) analysis that leverages tissue-specific gene expression regulatory networks to identify genetic effects on expression regulatory activities. Activity QTLs have been previously reported between master regulators (MRs) and functional coding or promoter variants in regulatory genes upstream of the MRs in their respective tissue-specific gene regulatory networks [11–13]. However, our present study is the first to explore the utility of systematically applying aQTL analyses for the elucidation of the molecular mechanisms mediating GWAS signals. These analyses have suggested candidate mediating genes for several cardiometabolic GWAS loci that to our knowledge do not yet have explanatory models or colocalizing *cis*-eQTLs.

## Results

### Inference of gene regulator activities from transcriptomic data

Based solely on transcriptomic data, gene activity inferences are restricted to genes whose products play a direct or indirect role in determining the gene expression landscape in the studied tissue. We therefore designated transcription factors, transcription co-factors and signal transduction factors as expression regulators (6,153 genes listed in **S1 Table**). An adipose gene co-expression network was inferred from 766 subcutaneous adipose RNA-seq samples from the TwinsUK Study using ARACNe (**S2 Table**) [14–17]. Network edges in ARACNe co-expression networks indicate significant mutual information (MI) between the two genes' expression. Our adipose ARACNe network contained 4,221 regulators and 13,775 targets with a total of 730,059 directed edges (**S2 Table**).

Regulator activities were then inferred using the VIPER R package based upon the adipose ARACNe co-expression network and the expression of the downstream targets in all 766 adipose RNA-seq samples [18]. Each target gene's influence on the inferred activity of its regulator is weighted by the significance of the association in the ARACNe network, is consistent with the direction of their Spearman correlation (i.e. the mode of regulation) and is adjusted by a pleiotropy correction to account for multiple upstream regulators [18]. Furthermore, activities were only inferred for regulators with at least 25 expressed targets to ensure the robustness of each inference.

### Master regulator (MR) analyses for cardiometabolic phenotypes in subcutaneous adipose tissue

Within this study we defined "master regulators" (MRs) as expression regulators that likely play critical roles in the gene regulatory program that instantiates a cell state associated with a given phenotype. These candidate MRs were identified as the regulators whose activities best

predict the phenotype of interest. We identified putative phenotypic MRs for subcutaneous adipose tissue by random forest regression modeling on regulator activities via the following steps: 1) test cross-validation error for random forest models with increasing numbers of regulators to determine a parsimonious number of regulators sufficient to minimize prediction error; 2) identify the most important regulators as measured by the percent increase in the mean square error (MSE) upon permutation; 3) train the final random forest model with the determined number of top regulators by importance (**S1 Fig**). Training and test sets were formed as a 70:30 split, respectively, of the TwinsUK adipose regulator activity profiles. Master regulators were identified for body mass index (BMI), waist-hip ratio (WHR), the natural log of the homeostatic model assessment of insulin resistance (HOMA-IR), plasma high-density lipids (HDL) and plasma triglycerides.

The cross-validation random forest analyses suggested ~100 regulators sufficiently minimized prediction error in the training sets for BMI, WHR, HOMA-IR and HDL, but triglycerides only needed ~60 regulators (**S2–S6** **Figs**). For each phenotype, the predictions by the final random forest models (with the indicated number of MRs selected by importance) were well correlated with actual measurements in both the training and test sets ($-\log_{10}(P)$ between 7.3 and 78.6; **Fig 1A–1O**). To further test these putative phenotypic MRs in an independent dataset we used the METSIM expression array data that includes 769 subcutaneous adipose samples from Finnish men [19]. First, regulator activities were inferred for the METSIM samples from the expression data and the TwinsUK adipose co-expression network using VIPER. Again, the associations between the actual phenotypic measurements and those predicted with the TwinsUK MR random forest models were highly significant for all phenotypes (**Fig 1C, 1F, 1I, 1L and 1O**). However, there were differences in phenotype distributions between the TwinsUK and METSIM datasets, which may in part be explained by gender differences (TwinsUK subjects are all female).

While correlations were noted between candidate MRs at the expression level, they were much stronger between activities (**S7A and S7B Fig**). This is expected as the MRs are selected based on activities rather than expression levels. Correlation between expression and activity for matched MRs was strong overall (mean r = 0.72), though there are some MRs with relatively weak correlation (i.e., r < 0.5), which highlights the important difference between these two metrics (**S7C and S7D Fig**). However, the matched MR correlations were clearly stronger than the unmatched (**S7D and S7E Fig**). The lack of clustering among MRs of the same phenotype is also notable, though perhaps not surprising for candidate MRs of well correlated cardiometabolic phenotypes (**S7A–S7C Fig**). This was also reflected in the overlap in MRs between phenotypes (**S4 Table** and **S7F Fig**).

## Activity QTL (aQTL) and eQTL analyses at cardiometabolic GWAS loci

The main question of this study was whether activity QTL (aQTL) analysis may provide a useful method to supplement eQTL analysis in identifying genes potentially mediating genetic associations with complex traits. Therefore, we restricted the eQTL and aQTL analyses to variants with genome-wide significant (i.e. $P \leq 5 \times 10^{-8}$, the conventional GWAS threshold) associations with cardiometabolic traits that may be in part mediated by adipose tissue. Specifically, we separately ran *cis*-eQTL and aQTL analyses for variants associated with BMI, BMI-adjusted WHR, BMI-adjusted type 2 diabetes (T2D), plasma HDL levels and plasma triglycerides levels against all nearby regulator genes (within 1Mb) for which activities were inferred. Note that these eQTL and aQTL results come from an all-female dataset and therefore specific associations cannot be safely generalized to both genders without further validation.

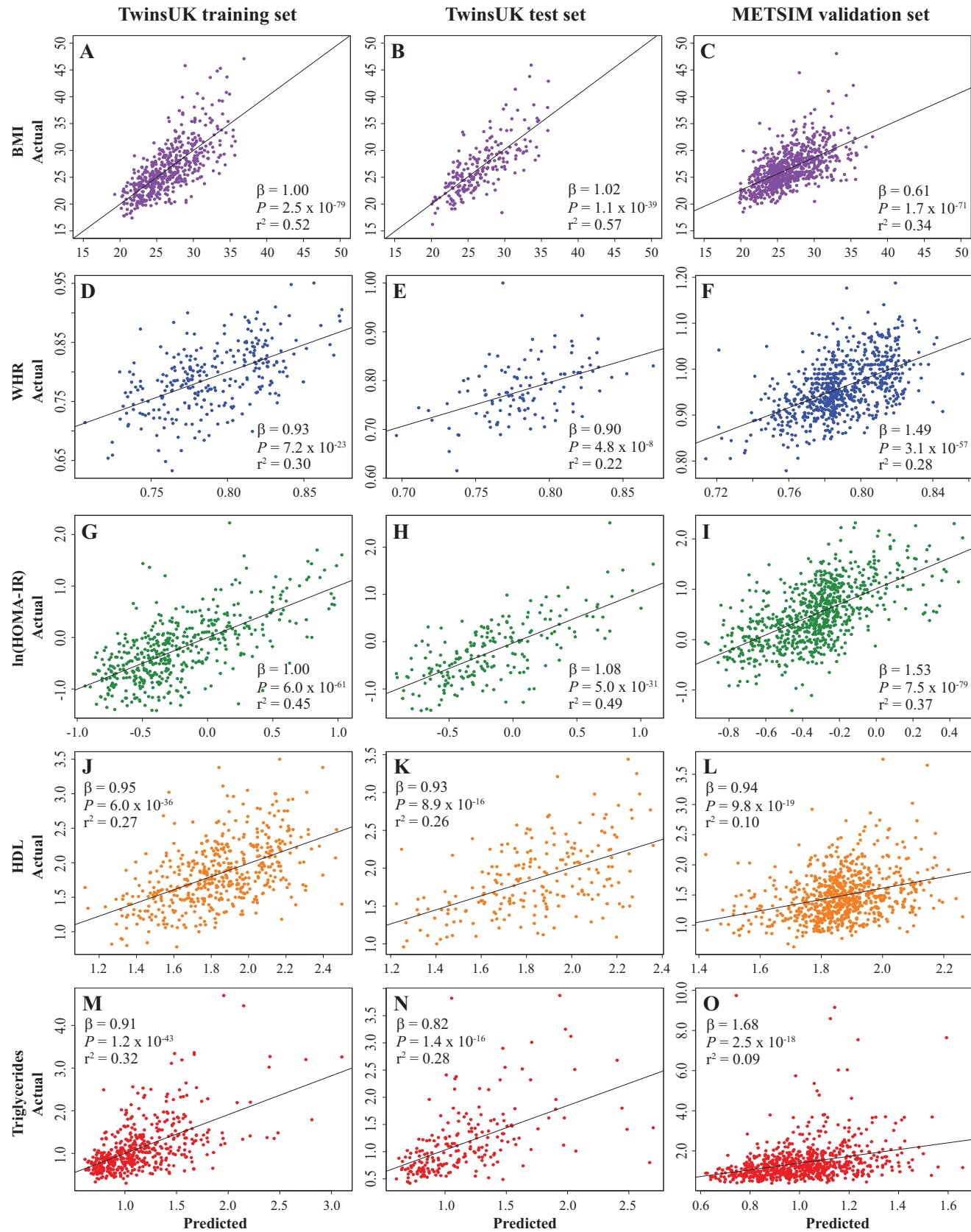

**Fig 1. Identification of phenotypic master regulators (MRs) in adipose tissue through random forest modeling. (A-O)** Scatter plots with trendlines comparing predicted and actual phenotypes for the TwinsUK training set (**A, D, G, J, M**), test set (**B, E, H, K, N**) and METSIM validation set (**C, F, I, L, O**) for BMI (**A-C**), WHR (**D-F**), HOMA-IR, (**G-I**), HDL (**J-L**) and triglycerides (**M-O**).

Overall, 55,855 GWAS variants were tested for association with the expression and activities of 4,213 expression regulators in the *cis* analyses and 291 candidate MRs in the *trans* analyses. These analyses together identified 264 *cis*-eRegulators, 11 *cis*-aRegulators, 20 *trans*-eMRs and 84 *trans*-aMRs significant at FDR < 0.05. In *cis*, eQTLs were almost always stronger than the respective aQTLs (**S5**–**S9 Tables**). This was expected since transcript-level effects are presumably more proximal to a functional variant's allelic effects in the chain of molecular events than gene product activities. However, among the 11 *cis*-aRegulators significant at FDR < 0.05 (versus 264 such *cis*-eRegulators), six aQTL signals had nominally better minimum *P*-values compared to their corresponding eQTL signals (**S10 Table**). Of those, only two were decisively so (nearly 2-orders of magnitude more significant than their corresponding eQTL signals): the *EPHB2* and *USP6 cis*-aQTL signals at the 1p36.1 and 17p13.2 BMI GWAS loci, respectively. *EPHB2* was the only significant aRegulator that was not also a significant eRegulator.

Given that MRs integrate upstream genetic and environmental information into a gene regulatory program that instantiates the cell state for a given phenotype, we hypothesized that expression levels and activities of putative phenotypic MRs may serve as sensitive gauges of the mediating molecular effects of GWAS variants (for matching or related phenotypes) when the given tissue is relevant to that particular GWAS signal. As such, by restricting *trans*-eQTL and aQTL analyses of GWAS signals to candidate MRs of matching or relevant phenotypes, we reduce the multiple testing burden of the analyses while simultaneously enriching the tested genes for those regulators most important to the phenotype of interest in the given tissue. We therefore assessed *trans*-eQTLs and aQTLs for significant BMI variants against BMI MRs, WHR (BMI-adjusted) variants against WHR MRs, T2D (BMI-adjusted) variants against HOMA-IR MRs, HDL variants against HDL MRs and triglycerides variants against triglycerides MRs. In contrast to the *cis* analyses, *trans*-aQTLs tended to be more significant than their respective eQTLs, especially for BMI and WHR GWAS signals (**Tables 1** and **S11**–**S15**). Among all unique QTLs for significant eRegulators/aRegulators (FDR < 0.05) across all analyses, only 20% of *cis*-aQTLs were more significant than their respective eQTLs while 72% of *trans*-aQTLs were more significant than their respective eQTLs (Z-test of independent proportions $P = 3.03 \times 10^{-280}$).

To ensure this aQTL advantage over eQTL in *trans* was not due to bias introduced by using activity scores rather than expression values to infer MRs, we also performed the random forest MR approach described above to identify the same number of putative phenotypic MRs based on expression. *Trans*-eQTL/aQTL analyses of these expression-based candidate MRs indicated that 82% of *trans*-aQTLs were more significant than their respective eQTLs among all unique, significant (FDR < 0.05) *trans*-QTLs. Therefore, *trans*-aQTLs maintain their advantage over *trans*-eQTLs regardless of the metric used to infer putative phenotypic MRs. However, activity score is expected to be a more robust metric preferred for identifying putative MRs (see Discussion) and using the expression-based MRs reduced the number of unique, significant *trans*-QTLs (778 for expression-based MRs versus 804 for activity-based MRs), so we proceeded in our analyses exclusively with the activity-based MR *trans*-QTLs.

## Regulators and master regulators tend to be under complex expression control

Recently, Wang and Goldstein reported a new Enhancer-Domain Score (EDS) that reflects the size, redundancy and conservation of all enhancers linked to a gene's expression [20]. They

**Table 1. Cardiometabolic GWAS loci with significant master regulator (MR) *trans*-QTLs.**

| GWAS | Locus | Best locus SNP | Chr | Position | Best locus *P* | MR trait | Significant MR *trans*-Genes | Significant MR *trans*-eGenes | Significant MR *trans*-aGenes |
|---|---|---|---|---|---|---|---|---|---|
| BMI | 1p36.1 | rs6692586 | 1 | 23299906 | 1.10E-16 | BMI | 41 | 4 | 41 |
| BMI | 1q24.3 | rs16864515 | 1 | 171435542 | 1.70E-10 | BMI | 4 | 0 | 4 |
| BMI-adjusted WHR | 1q24.3 | rs714515 | 1 | 172352990 | 4.40E-15 | WHR | 44 | 1 | 44 |
| BMI | 1q25.2 | rs543874 | 1 | 177889480 | 1.20E-122 | BMI | 1 | 0 | 1 |
| BMI-adjusted WHR | 1q41 | rs2820443 | 1 | 219753509 | 5.30E-21 | WHR | 4 | 0 | 4 |
| Triglycerides | 2p24.1 | rs676210 | 2 | 21231524 | 3.28E-71 | Triglycerides | 1 | 1 | 0 |
| BMI | 2p23.3 | rs12468863 | 2 | 26940294 | 5.10E-21 | BMI | 1 | 1 | 0 |
| BMI-adjusted WHR | 2q24.3 | rs1128249 | 2 | 165528624 | 2.00E-15 | WHR | 1 | 0 | 1 |
| HDL | 3p25.3 | rs2606736 | 3 | 11400249 | 4.80E-08 | HDL | 1 | 1 | 0 |
| BMI | 5q31.2 | rs7716275 | 5 | 137631073 | 2.20E-10 | BMI | 1 | 0 | 1 |
| BMI-adjusted WHR | 6q22.3 | rs1936805 | 6 | 127452116 | 3.60E-35 | WHR | 2 | 1 | 1 |
| Triglycerides | 7q32.2 | rs287621 | 7 | 130435181 | 7.67E-09 | Triglycerides | 3 | 3 | 0 |
| HDL | 7q32.2 | rs11765979 | 7 | 130445877 | 3.11E-17 | HDL | 3 | 3 | 0 |
| BMI | 7q32.2 | rs972283 | 7 | 130466854 | 5.10E-09 | BMI | 1 | 1 | 0 |
| BMI-adjusted T2D | 7q32.2 | rs61462211 | 7 | 130468015 | 1.00E-16 | HOMA-IR | 2 | 2 | 0 |
| BMI | 8q21.2 | rs733594 | 8 | 85077686 | 5.90E-14 | BMI | 3 | 1 | 2 |
| BMI | 10q26.3 | rs4880341 | 10 | 133992689 | 1.10E-11 | BMI | 3 | 0 | 3 |
| BMI | 11q13.1 | rs7102454 | 11 | 65594820 | 2.4E-18 | BMI | 3 | 0 | 3 |
| Triglycerides | 11q23.3 | rs10790162 | 11 | 116639104 | 1.1E-249 | Triglycerides | 1 | 1 | 0 |
| BMI | 12p13.33 | rs11611246 | 12 | 939480 | 5.00E-32 | BMI | 6 | 0 | 6 |
| BMI | 12p13.1 | rs12422552 | 12 | 14413931 | 1.60E-11 | BMI | 8 | 0 | 8 |
| BMI | 12q13.13 | rs4759075 | 12 | 54667285 | 1.40E-11 | BMI | 1 | 0 | 1 |
| HDL | 15q21.3 | rs10468017 | 15 | 58678512 | 1.21E-188 | HDL | 1 | 1 | 0 |
| BMI | 15q24.1 | rs7164727 | 15 | 73093991 | 3.30E-25 | BMI | 3 | 0 | 3 |
| HDL | 17q25.3 | rs4969178 | 17 | 76388202 | 1.53E-12 | HDL | 1 | 1 | 0 |
| BMI | 18q21.3 | rs663129 | 18 | 57838401 | 1.60E-178 | BMI | 1 | 0 | 1 |
| BMI | 19q13.3 | rs3810291 | 19 | 47569003 | 2.10E-52 | BMI | 1 | 1 | 0 |
| BMI-adjusted T2D | 20q13.32 | rs736266 | 20 | 57387352 | 1.00E-11 | HOMA-IR | 1 | 1 | 0 |

Table includes all tested cardiometabolic GWAS loci with at least one significant MR *trans*-QTL at FDR < 0.05. See S11–S15 Tables for all significant MR *trans*-eQTLs and aQTLs at tested GWAS loci.

found that genes involved in development or pathogenicity and those with nearby GWAS signals tended to have higher EDS, while in contrast, significant *cis*-eGenes (in GTEx across all tissues) tended to have lower EDS. This mismatch between EDS distributions for GWAS genes and significant *cis*-eGenes is concerning for the prospect of identifying the genes mediating GWAS signal effects via *cis*-eQTL analysis alone. Therefore, we decided to explore the EDS distributions among significant aGenes, MRs and *trans*-Genes (**S16 Table**).

First, we noted that expression regulators in general have significantly higher mean EDS compared to all expressed genes in the TwinsUK data ($P_{t-test} = 1.20 \times 10^{-32}$; **Fig 2** and **S16 Table**). We also observed lower mean EDS for *cis*-eGenes (those with an eQTL $P < 5 \times 10^{-6}$) compared to all expressed genes and for *cis*-eRegulators (those with an eQTL $P < 5 \times 10^{-6}$)

compared to all expressed regulators (respectively, $P_{t-test}$ = 9.38 x $10^{-5}$ and $P_{t-test}$ = 0.0022; **Fig 2A** and **S16 Table**). In contrast, *cis*-aRegulators (those with an aQTL $P < 5$ x $10^{-6}$) have a higher mean EDS than all expressed regulators, though the difference was not significant, perhaps due to a lack of power (**Fig 2A** and **S16 Table**). Unlike *cis*-eGenes, *trans*-eGenes (those with an eQTL $P < 1$ x $10^{-8}$) had a significantly higher mean EDS than all expressed genes ($P_{t-test}$ = 7.57 x $10^{-4}$; **Fig 2B** and **S16 Table**), though their mean was still significantly lower than for all expressed regulators ($P_{t-test}$ = 0.0060; **Fig 2B** and **S16 Table**). Both *trans*-eRegulators and *trans*-aRegulators (those with an eQTL $P < 1$ x $10^{-8}$) had significantly higher mean EDS than all expressed regulators (respectively, $P_{t-test}$ = 0.0097 and $P_{t-test}$ = 1.22 x $10^{-4}$; **Fig 2B** and **S16 Table**). The mean EDS scores of the *trans*-eMRs and *trans*-aMRs identified above with GWAS variants were even higher than the *trans*-eRegulators and *trans*-aRegulators (*trans*-eMRs vs. *trans*-eReglators $P_{t-test}$ = 0.0956 and *trans*-aMRs vs. *trans*-aReglators $P_{t-test}$ = 0.0468; **Fig 2B** and **S16 Table**).

Given the consistent trend of higher mean EDS among genes identified by aQTL versus eQTL, it is tempting to speculate that aQTLs have an advantage in detecting genes with more complex enhancer structure, which are enriched near GWAS signals. However, given the numbers of detected aGenes in these analyses, larger sample sets are needed to reach a firm conclusion on this. What is clear is that expression regulator genes and significant *trans*-eGenes trend towards higher EDS than genes in general, and likewise for *trans*-e/aRegulators and *trans*-e/aMRs compared to expression regulators in general. This seems to imply that genes in general, and expression regulator genes specifically, may be more likely to have detectable *trans*-QTL associations when they are regulated by a more complex enhancer structure.

## Further analyses of select GWAS loci

The *trans*-QTL results presented in **Table 1** represent cases in which *trans*-eQTLs/aQTLs (for the putative phenotypic MRs inferred from regulatory activities) suggest hypotheses for the mediation of GWAS signals that may warrant follow-up functional studies. This is progress, since for most of these GWAS signals, standard eQTL analyses have failed to suggest genes that mediate their effects. However, mere overlap between a GWAS signal and e/aQTL signal is relatively weak evidence that the latter mediates the former as the two signals may result from distinct functional variants with incidental correlation between the two signals [7]. We therefore used the HyPrColoc R package to perform formal colocalization analysis between the GWAS signals and all QTL signals for *cis* genes and relevant *trans* MRs at the 24 loci listed in **Table 1** as well as the 17p13.2 locus that contained a BMI GWAS signal that overlapped the *USP6 cis*-aQTL signal mention previously. At a posterior probability (PP) >0.5, 13 *cis*-eQTL, 2 *cis*-aQTL, 55 *trans*-eQTL and 166 *trans*-aQTL signals colocalized with GWAS signals (full results in **S17 Table**).

The chr7q32.2 locus, which includes GWAS signals for BMI, T2D, HDL and triglycerides, has previously been studied in some detail (**Fig 3A**) [8,9]. Small *et al*. reported an adipose-specific *KLF14 cis*-eQTL signal overlapping the T2D and HDL GWAS signals, as well as 385 *trans*-eQTLs with genes enriched in "metabolic processes" and "binding by PPARG and RXRA during adipocyte differentiation" GO: Biological Processes [21,22]. Therefore, we assessed the results of our analyses at the 7q32.2 locus against those of Small *et al*. and performed colocalization analyses for our eQTL and aQTL results with BMI, T2D (BMI-adjusted), HDL and triglycerides GWAS signals using HyPrColoc [23]. We observed significant *KLF14 cis*-eQTLs and *cis*-aQTLs overlapping the BMI, T2D (BMI-adjusted), HDL and triglycerides GWAS signals, though the eQTL signal was more significant (**Fig 3B**; eQTL

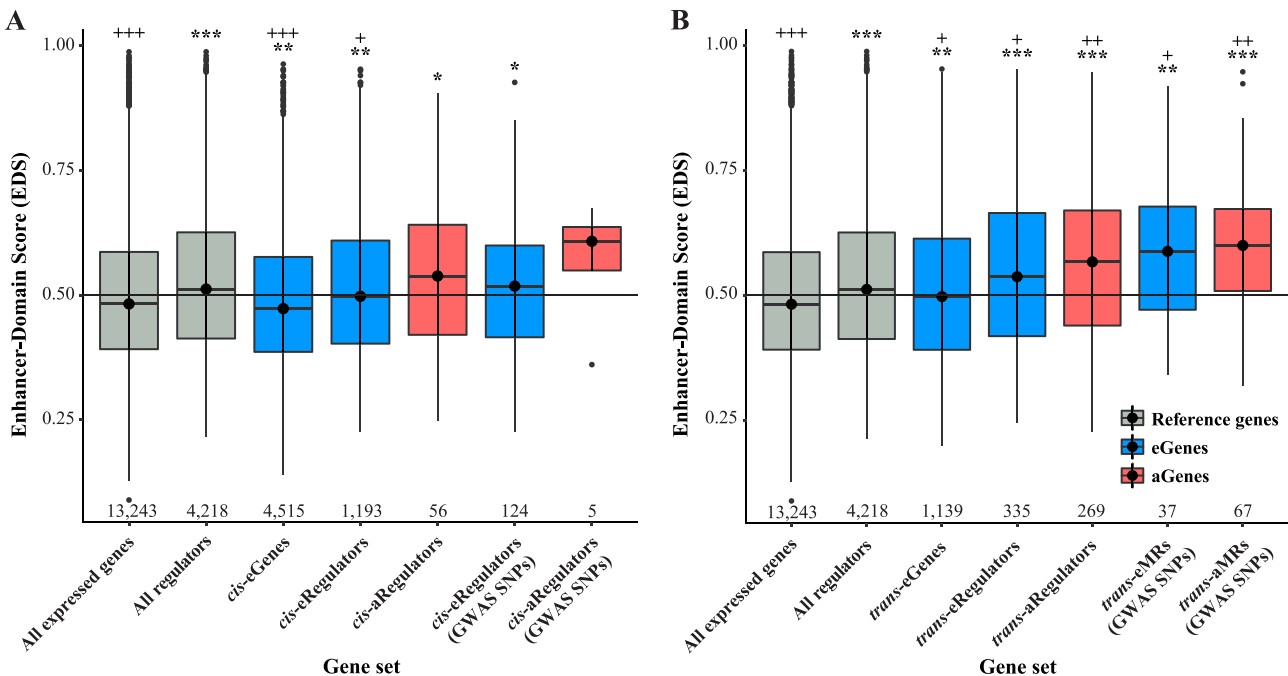

**Fig 2. EDS score distributions among genes and regulators with *cis* or *trans*-QTLs.** Box plots of the distribution of Enhancer-Domain Scores (EDS) among (**A**) *cis*-eGenes (with $P_{min} < 5 \times 10^{-6}$ among all SNPs and *cis*-Genes) and *cis*-e/aRegulators (with $P_{min} < 5 \times 10^{-6}$ among all SNPs or restricted to GWAS SNPs and *cis*-Regulators) or (**B**) *trans*-eGenes (with $P_{min} < 1 \times 10^{-8}$ among all SNPs and genes), *trans*-e/aRegulators (with $P_{min} < 1 \times 10^{-8}$ among all SNPs and expression regulators) and e/aMRs (with $P_{min} < 5 \times 10^{-6}$ among GWAS SNPs and relevant MRs). For both panels, t-test was used to assess significance of difference in mean EDS between the indicated gene set and all expressed genes ($P = 0.05$ for *, $P = 0.0005$ for **, $P = 5 \times 10^{-6}$ for ***), or between the indicated gene set and all expressed regulators ($P = 0.05$ for +, $P = 0.0005$ for ++, $P = 5 \times 10^{-6}$ for +++). The numbers along the bottom of each plot indicate the gene counts for the set.

$P_{rs972283} = 4.14 \times 10^{-29}$ vs. aQTL $P_{rs972283} = 2.50 \times 10^{-5}$). Furthermore, significant *trans*-eQTLs (at FDR < 0.05) for six MRs (*AGT*, *GNB1*, *RABIF*, *NR2F1*, *ESR2* and *TBX4*) also overlapped the GWAS signals (**Tables 1, S11, and S13–S15**). The four GWAS signals did colocalize well (*PP* > 0.5) with each other, though the BMI-HDL and T2D-triglycerides pairs colocalized better than other pairings (**S8 Fig** and **S17 Table**). Plots comparing the SNP-wise -$\log_{10}(P)$ (i.e. LocusCompare plot) for each pairing of the four GWAS suggested this may be due to at least two functional signals represented by rs972283 and rs287621, which have an $r^2 = 0.31$ in the 1000 Genomes EUR population (**Figs 3A, 3C–3J** and **S8**). While the BMI and HDL signals appear best explained by the rs972283 signal, the triglycerides signal appears better explained by the rs287621 signal and the T2D signal appears to include both signals at roughly equal strength (**S8 Fig**). Conditional analyses of the GWAS summary statistics with COJO were consistent (though not definitive) with the presence of the two functional signals for T2D, but not for the other three GWAS (**S18–S21 Tables**) [24,25]. This impacted the colocalization analyses as the presence of two T2D signals necessarily reduced the posterior probability (*PP*) that there is a single shared functional variant with any eQTLs and aQTLs. The BMI and HDL GWAS signals colocalized strongly with *LINC-PINT*, *AC016831.7* and *KLF14* *cis*-eQTLs, *KLF14* *cis*-aQTL and several MR *trans*-eQTLs and aQTLs (*PP*$_{median}$ = 0.89 [range 0.48–0.97]; **Figs 3C, 3E, 3G, 3I** and **S9**, and **S17 Table**). In contrast, colocalizations tended to be much weaker for T2D and triglycerides GWAS signals with the various *cis* and *trans* eQTL and aQTL signals (*PP*$_{median}$ = 0.59 [range 0.41–0.97]; **Figs 3D, 3F, 3H, 3J** and **S9**, and **S17 Table**). Notably, the only *cis* signals to colocalize with the T2D signal were the *AC016831.7* eQTL (PP = 0.62) and the *KLF14* aQTL (PP = 0.43) (**Figs 3H** and **S9**, and **S17 Table**). Also, all MRs that had a *trans*-

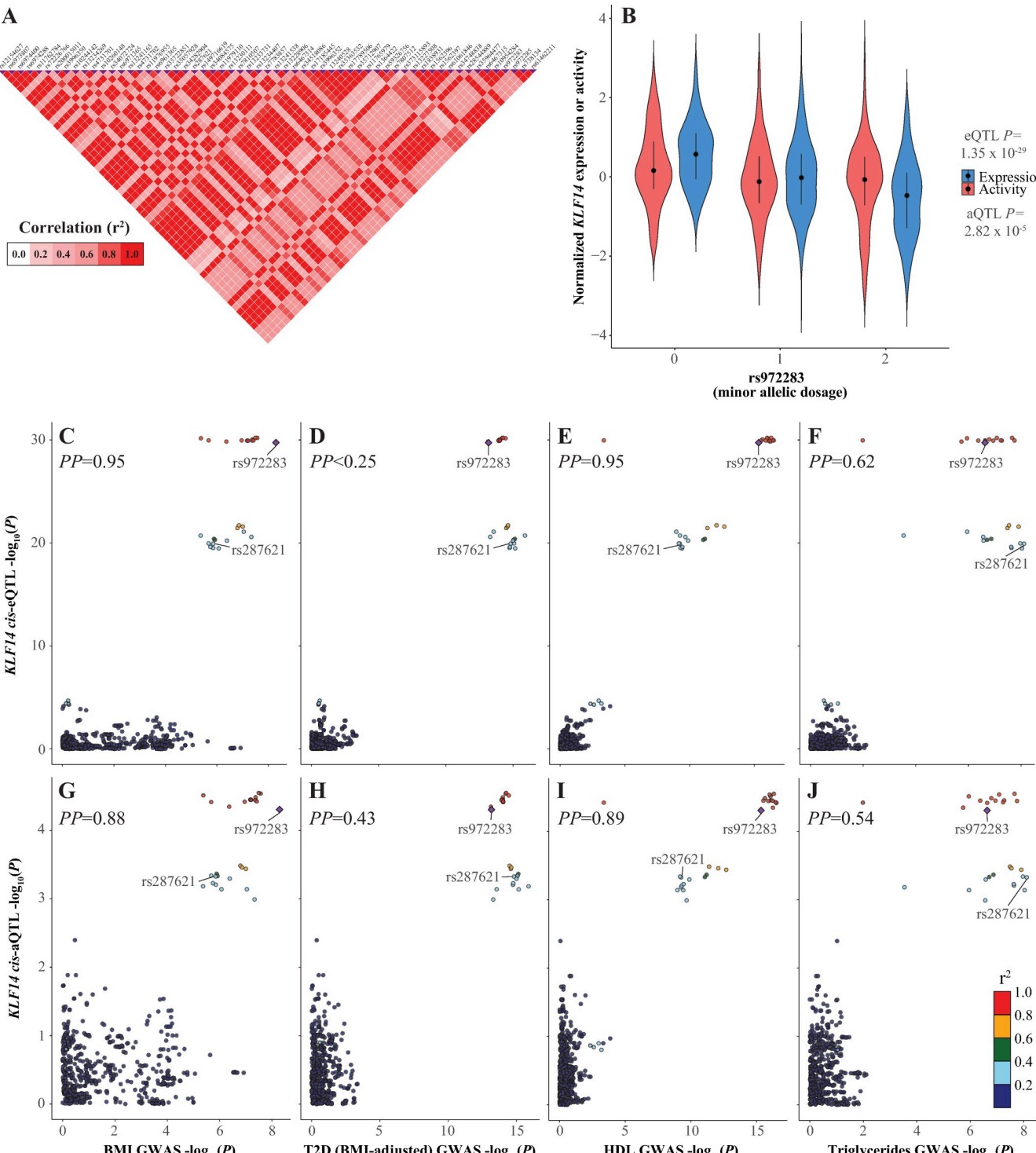

**Fig 3.** ***KLF14 cis*-eQTL/aQTL signals and their colocalization with GWAS signals at chr7q32.2.** **(A)** LD matrix shaded by correlation ($r^2$) for chr7q32.2 GWAS variants associated with BMI, T2D (BMI-adjusted), HDL and triglycerides. **(B)** Violin plot of normalized *KLF14* expression or activity distributions stratified by rs972283 genotype. **(C-J)** LocusCompare plots comparing the $-\log_{10}(P)$ for variants from the indicated GWAS signals versus *KLF14 cis*-eQTL or aQTL signals. Posterior probabilities (*PP*) of a single, common functional variant for the compared association signals were calculated with HyPrColoc.

eQTL or aQTL that colocalized with the GWAS signals were also identified in *trans*-eQTLs by Small *et al.*, which they had validated in other datasets (**S17 Table**) [9].

In contrast to the complexity of the pleiotropic 7q32.2 locus, the BMI locus at chr12p13.1 has only one likely functional candidate variant, rs12422552 (**Figs 4** and **S10**). Strikingly, this BMI signal colocalizes with 15 BMI MR *trans*-eQTLs ($PP_{median}$ = 0.67 [range 0.49–0.98]) and 64 BMI MR *trans*-aQTLs ($PP_{median}$ = 0.89 [range 0.49–1.00]), though among these nominally significant, colocalized *trans*-QTL signals, only 8 *trans*-aQTLs (*ANG*, *CSNK2A2*, *ID2*, *PIM1*, *PTPRJ*, *TENM4*, *TNFRSF10C* and *ZFAT*) were significant at FDR < 0.05 (**Fig 4**, **and Tables 1**, **S11**–**S15** and **S17**). Despite the many colocalized *trans* effects on BMI MR activities, no *cis*-eQTL or *cis*-aQTL colocalized with the 12p13.1 BMI GWAS signal. This locus was just one example among several in this study (1q24.3 BMI, 1q24.3 WHR, 3p25.3 HDL, 5q31.2 BMI, 10q26.3 BMI, 17q25.3 HDL, 20q13.32 T2D) where colocalizing MR *trans*-eQTLs/aQTLs were identified in the absence of any colocalizing *cis*-eQTLs/aQTLs, and among such loci, *trans*-aQTLs were typically stronger than their cognate eQTLs (**S11**–**S15** and **S17 Tables**).

The BMI GWAS locus at 1p36.1 was rather exceptional among the loci we analyzed in that it contains a *cis*-aQTL signal (for *EPHB2*) that is stronger than its cognate eQTL signal while also having 41 BMI MR *trans*-aQTLs significant at FDR < 0.05 (**Figs 5, 6C and 6D**, **Tables 1**, **S5 and S11**). All 41 FDR-significant *trans*-aQTLs were stronger than their corresponding *trans*-eQTLs and this trend also extended to the nominally significant *trans*-QTLs (**Fig 6A and 6B** and **S11 Table**). Notably, *EPHB2* was also identified as a putative BMI MR (**S4 Table**). The *EPHB2 cis*-aQTL signal was the only *cis*-QTL to colocalize with the BMI GWAS signal ($PP$ = 0.86). Furthermore, 35 nominally significant BMI MR *trans*-aQTLs colocalized ($PP$ > 0.25) with the BMI signal, versus only 5 *trans*-eQTLs (**S17 Table**). Interestingly, Locus-Compare plots for the BMI GWAS and each QTL again suggested at least two functional signals at this locus represented by rs6692586 and rs4654828 ($r^2$ = 0.37 in the 1000 Genomes EUR population) that were distinguishable to varying degrees among the tested QTLs at this locus, though conditional analysis of the BMI GWAS summary statistics with COJO was indecisive on the independence of BMI effects for these SNPs (**Fig 5A and 5C–5I** and **S22 Table**). For example, the rs6692586 signal was clearly the strongest, if not the sole, signal for the BMI GWAS, while for the *DOK5 trans*-aQTL the rs4654828 signal was stronger, and for the *EPHB2 cis*-aQTL the two signals were roughly equivalent (**Fig 5C and 5D**). Consequently, there were many BMI MR *trans*-QTLs that colocalized only weakly ($0.25 < PP < 0.50$) with the BMI GWAS signal, but colocalized well ($PP > 0.50$) with the *EPHB2 cis*-aQTL, which itself colocalized well with the BMI GWAS signal (**Fig 5C–5I** and **S17 Table**). In the adipose co-expression network *EPHB2* was directly connected to four other BMI MRs that had significant *trans*-aQTLs that colocalized with the BMI GWAS signal, and it was a 2nd degree neighbor of all other such BMI MRs (**Fig 6**). Therefore, while the presence of two putative functional signals at this locus complicates the interpretation of the BMI GWAS signal, the aQTL results implicated *EPHB2* as a potential mediator of the genetic BMI effects of this locus in adipose tissue via its downstream effects on the adipose gene regulatory program associated with BMI.

## Discussion

In this study we have attempted to leverage information about the tissue-specific gene expression regulatory landscape to better inform hypothesis generation about the molecular consequences of genetic signals that mediate their effects on complex traits. Toward this end, we inferred activity scores for expression regulators that are distillations of patterns in the expression levels of their downstream targets in the tissue-specific regulatory network. These inferred

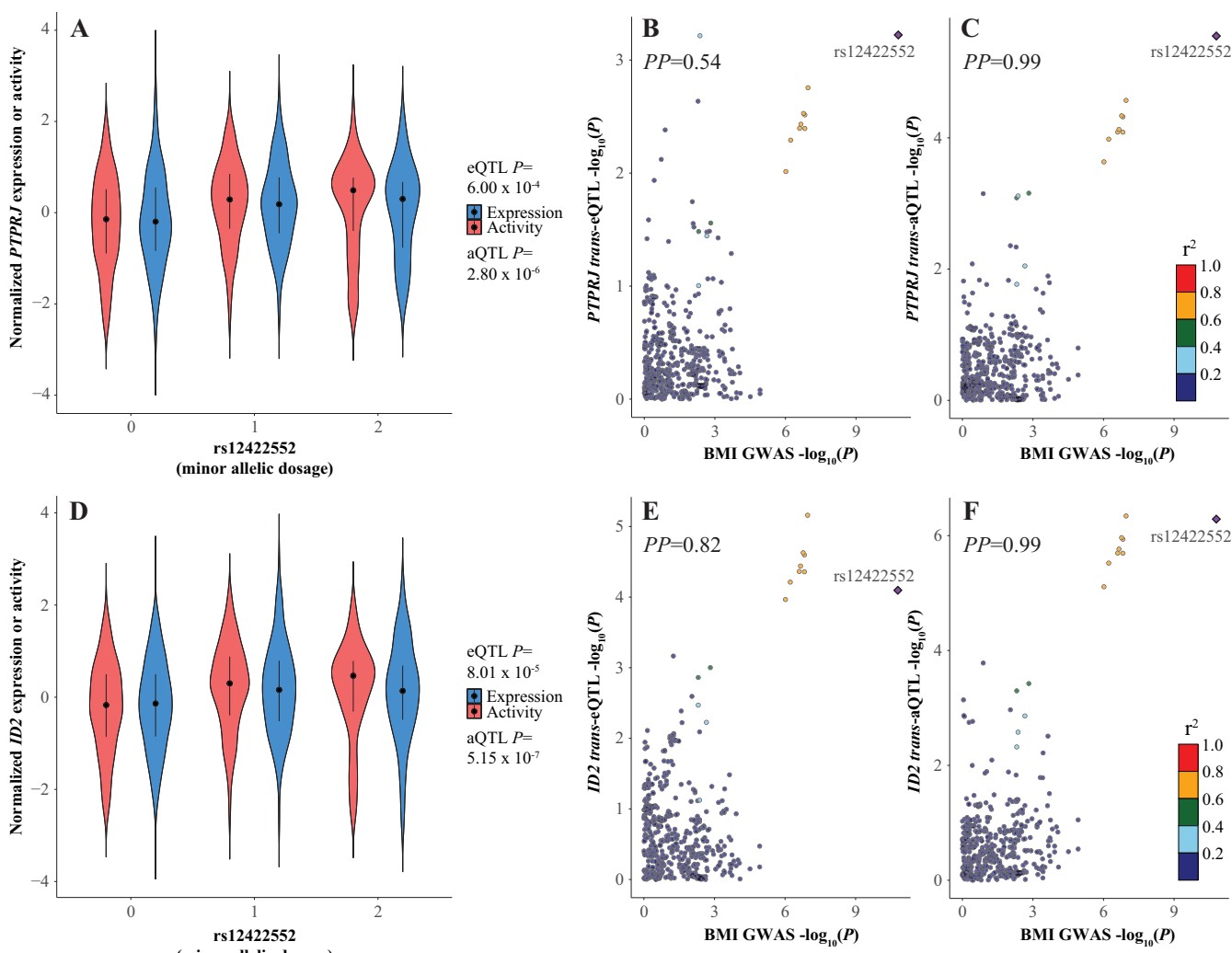

**Fig 4. *PTPRJ* and *ID2* BMI master regulators *trans*-eQTL/aQTL signals and their colocalization with the BMI GWAS signal at chr12p13.1. (A)** Violin plot of normalized *PTPRJ* expression or activity distributions stratified by rs12422552 genotype. **(B-C)** LocusCompare plots comparing the -log$_{10}$(*P*) for variants from the BMI GWAS signal versus the *PTPRJ trans*-eQTL or aQTL signals. Posterior probabilities (*PP*) of a single, common functional variant for the compared association signals were calculated with HyPrColoc. **(D)** As in **A**, but for *ID2* expression or activity distributions. **(E-F)** As in **C-D**, but for *ID2 trans*-eQTL or aQTL signals.

activities have several advantages and disadvantages as quantitative molecular traits as compared to transcript expression levels.

First, because the activity is dependent on the expression of many genes, it is far more robust to noise in the expression data [18]. Standard transcriptomic approaches measure steady-state expression as an average over many cells, which masks the complexity of expression dynamics within cells that are not necessarily synchronized across a tissue sample [26,27]. In contrast, inferred activity scores not only accommodate these unavoidable fluctuations due to the dynamics of transcription through their robustness but are able to capture coordinated, meaningful fluctuations in gene expression profiles. However, this and other advantages are completely dependent on the accuracy of the activity inference that is in turn dependent on the adequacy of the tissue/context-specific gene co-expression network. Also, using only

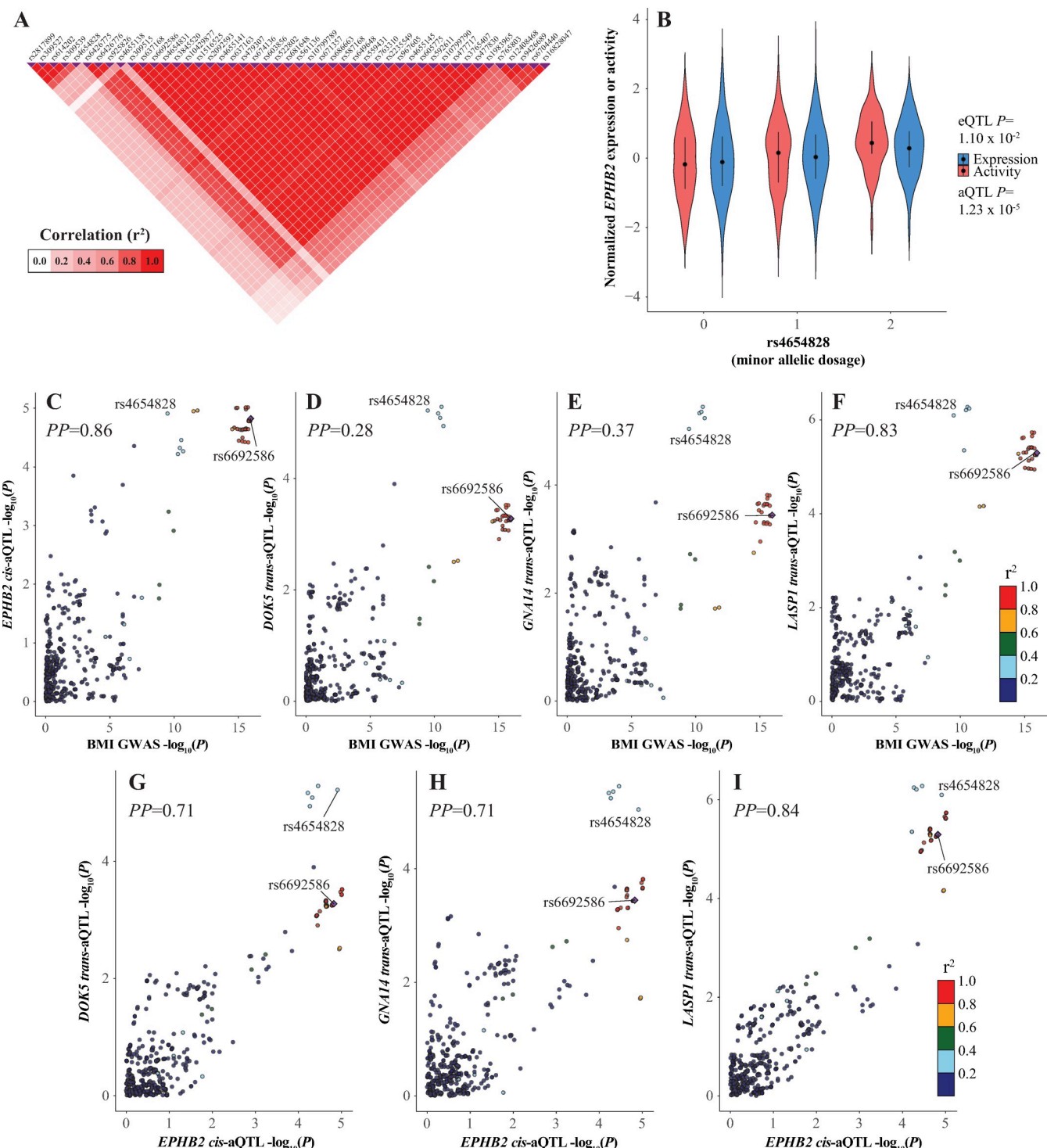

**Fig 5. *EPHB2 cis*-eQTL/aQTL and BMI master regulator *trans*-eQTL/aQTL signals and their colocalization with the BMI GWAS signal at chr1p36.1.**
(**A**) LD matrix shaded by correlation (r²) for chr1p36.1 BMI GWAS variants. (**B**) Violin plot of normalized *EPHB2* expression or activity distributions stratified by rs4654828 genotype. (**C-F**) LocusCompare plots comparing the -log₁₀(P) for variants from the BMI GWAS signal versus the *EPHB2 cis*-aQTL signal (**C**) or indicated BMI master regulator *trans*-aQTL signal (**D-F**). (**G-I**) LocusCompare plots comparing the -log₁₀(P) for variants from the *EPHB2 cis*-aQTL signal versus the indicated BMI master regulator *trans*-aQTL signal. Posterior probabilities (*PP*) of a single, common functional variant for the compared association signals were calculated with HyPrColoc.

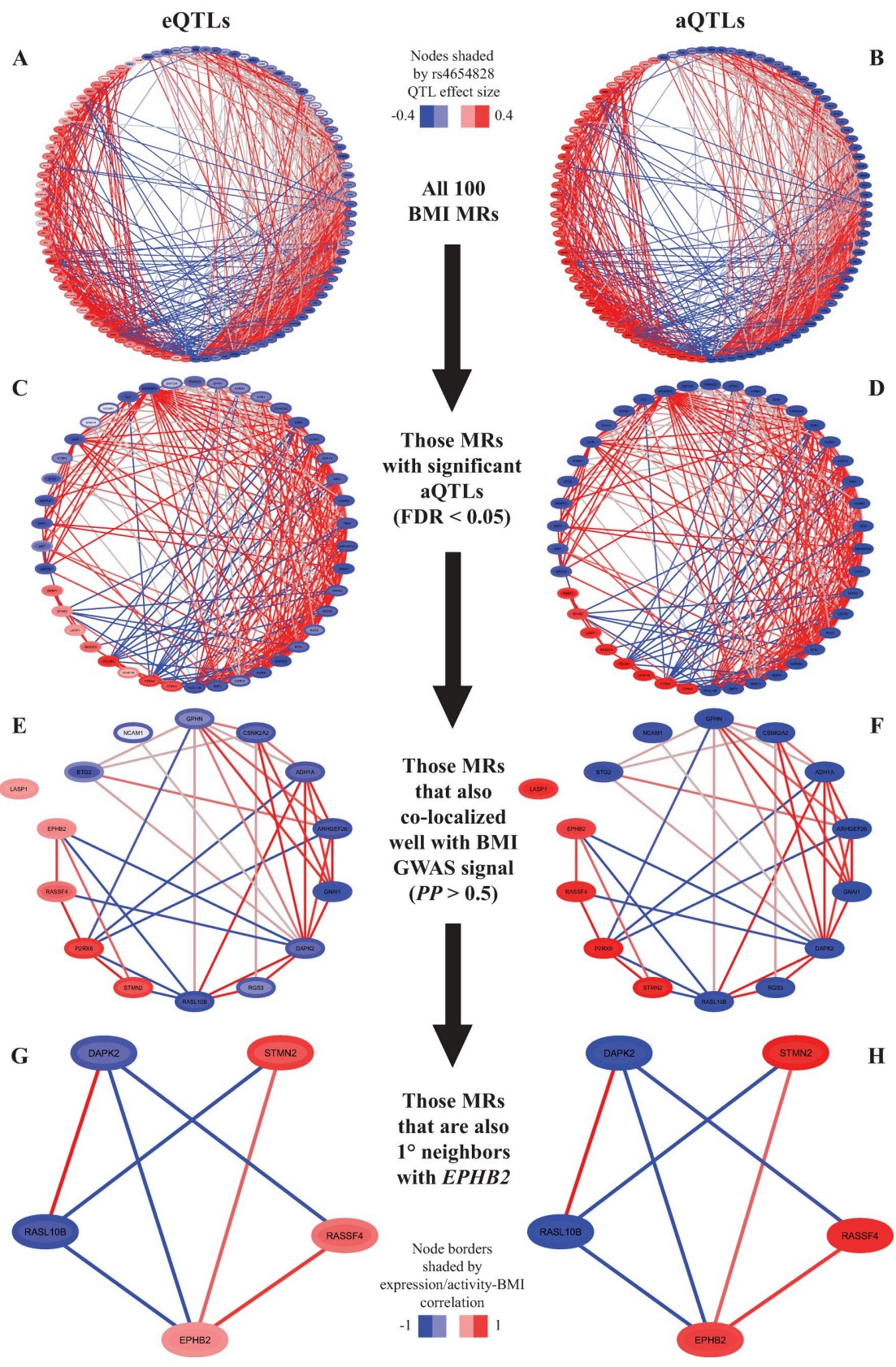

**Fig 6. Adipose co-expression subnetworks of BMI master regulators with associations at the 1p36.1 BMI GWAS locus. (A-B)** Subnetwork graph of all putative BMI master regulators. **(C-D)** Subnetwork graphs of BMI master regulators with a significant aQTL with rs4654828. **(E-F)** Subnetwork graph of BMI master regulators with a significant aQTL with rs4654828 that also colocalized well with the BMI GWAS signal ($PP > 0.50$). **(G-H)** Subnetwork graph of BMI master regulators with a significant aQTL with rs4654828 that also colocalized well with the BMI GWAS signal ($PP > 0.50$) and are first degree neighbors of *EPHB2*. All subnetworks were extracted from the full adipose co-expression network. For all graphs nodes are shaded by eQTL (**A,C,E,G**) or aQTL (**B,D,F,H**) effect sizes for rs4654828, node borders are shaded by Pearson correlation between BMI and the regulator's expression (**A,C,E,G**) or activity (**B,D,F,H**), and edges are shaded by the Spearman correlation (i.e. mode of action; red indicates positive correlation and blue indicates negative) between regulator and target. Note that Spearman correlations may be near 0 due to non-linear associations since edge inclusion was based on significant mutual information.

transcriptome data, activities may only be inferred for genes that influence expression profiles, like transcription factors (TFs), transcription co-factors and signal transduction factors.

Second, TF genes tend to be lowly expressed compared to other genes, which implies a smaller expected variance in transcript levels under normal circumstances [26,28]. This relatively low variance in the presence of the unavoidable noise inherent to steady-state transcriptomic approaches suggests lower statistical power to detect eQTLs for TF genes. In contrast, activity QTLs (aQTLs) would not face such limitations as they are derived from many downstream target genes with a range of expression means and variances. Though, as above, this advantage depends upon the adequacy of the tissue/context-specific gene co-expression network used for the activity inference.

Third, expression regulator activities are far closer in the causal chain to complex phenotypes of interest than transcript levels but are consequently further removed from the proximal molecular consequences of allelic variation. This larger distance in the causal chain between genotype and activity provides increased opportunity for noise to be amplified, likely decreasing the measured association between the two. Our comparison of *cis*-eQTLs and aQTLs bears out this expectation. However, the increased distance between genotype and activity in the causal chain also provides room for the canalization of the effects of genetic signals, potentially in concert with other genetic signals and environmental interactions, such that the downstream molecular consequences relevant to a complex trait of interest may become apparent. We hypothesized this may be best observed through *trans* effects on phenotypic master regulators (MRs) for a tissue that is relevant to the complex trait of interest. Indeed, we observed many significant MR *trans*-aQTLs at cardiometabolic GWAS loci, and these were often more significant than their corresponding *trans*-eQTLs. Furthermore, these *trans* effects on phenotypic MRs in relevant tissues are theoretically not restricted to the tissue wherein the proximal, allele-specific molecular effects occurred.

Finally, while many genetic signals associated with complex traits may be mediated by effects on the steady-state gene-level expression, genetic effects on other molecular phenotypes have been suggested as possible mediators of complex traits in the absence of detectable, gene-level eQTLs [29,30]. Examples include missense coding variants, genetic effects on RNA modifications (as in $m^6A$-QTLs) and isoform QTLs that indicate genetic effects on alternative splicing, promoter usage, or differential polyadenylation/cleavage sites [31–34]. Furthermore, there has been evidence that transcription kinetics of expression regulators can influence their activity without any observable change in steady-state transcript levels [27,35]. Activity QTLs are theoretically capable of capturing all these allele-specific molecular effects, though only for genes that encode expression regulators and only when such effects are relevant to the activity of the expression regulator. However, an aQTL does not reveal which of these upstream molecular effects might mediate the activity effect. Conversely, even if such upstream molecular effects (including eQTLs) are detectable for a given genetic signal, its relevance to other phenotypes would be suspect without a consequent detectable effect on the regulator's activity.

In this study we inferred regulator activities for sub-cutaneous adipose tissue samples from the TwinsUK Study using VIPER [16–18]. However, there is an updated version called meta-VIPER that infers activities based on multiple co-expression networks such that each regulator may use a different regulon (i.e., the subnetwork of a regulator with its downstream targets) based on the enrichment of differentially expressed target genes in the regulons across all input networks [36]. This is particularly useful when the cell type or context of a sample is uncertain or when an expression regulator's transcript levels have too little variance to detect correlation in the samples used to infer the context-specific co-expression network. Depending on the context being studied and the co-expression networks available, metaVIPER could further optimize the MR and aQTL approaches described here.

Another potential avenue for optimizing this approach would be the application of batch effect correction methods, like PEER factors, for the aQTL linear models [37]. Such methods have been shown to increase power in eQTL analyses [37]. However, the effects for which PEER factors adjust are unknown, and may correlate with phenotypes relevant to the analysis at hand if not included as covariates in the PEER training. Given the wide range of potentially relevant phenotypes under consideration in our study, including some for which we lacked data, we opted to simplify the analyses by omitting PEER factors from QTL models at the likely cost of some loss in statistical power. In future analyses, careful application of batch effect correction may further optimize the aQTL method.

Setting aside the above discussed advantages and limitations, this MR and aQTL approach still enjoys the benefit of requiring no additional data beyond the genotype, phenotype and expression data already available in existing eQTL datasets. As such, the approach may be considered low risk with potentially high reward. Indeed, our analyses did generate some intriguing hypotheses for molecular traits that may mediate the effects of various cardiometabolic GWAS signals, though it is again worth noting that these aQTL results have yet to be validated in a male dataset. Consistent with previous studies, we found that the BMI, T2D (BMI-adjusted), HDL and triglycerides GWAS signals at chr7q32.2 colocalize with *KLF14*, *LINC-PINT* and *AC016831.7 cis*-eQTLs, *KLF14 cis*-aQTL and several MR *trans*-eQTLs and aQTLs to varying degrees [8,9,38]. Though all these colocalized eQTLs were previously observed, the colocalization with a significant *KLF14 cis*-aQTL was a new observation that lends further weight to the hypothesis that *KLF14* expression and activity may mediate at least some component of these cardiometabolic GWAS signals. Conversely, the lack of significant *LINC-PINT cis*-aQTL casts doubt on its relevance to these GWAS signals. *AC016831.7* is uncharacterized and consequently excluded from the aQTL analysis, but it is notable that its *cis*-eQTL colocalized best with the T2D (BMI-adjusted) GWAS signal among all *cis*-QTLs. The generally weaker colocalizations with the T2D (BMI-adjusted) GWAS was likely dues the presence of two putative functional signals at the locus.

We also noted several examples of GWAS loci without any colocalized *cis*-QTLs that none-the-less exhibited significant, colocalized *trans*-QTLs with relevant phenotypic MRs (e.g., the 12p13.1 BMI locus). Furthermore, among the MR *trans*-QTLs tested, the MRs' aQTLs tended to be stronger than their eQTLs, and this effect was not due to any biasing from using activity scores to identify the putative MRs. This is particularly relevant given previous observations that *trans*-eQTLs are more strongly enriched among GWAS variants than *cis*-eQTLs, which may be related to observations by Wang and Goldstein, and this study, that both GWAS signals and *trans*-eGenes are enriched for high enhancer domain complexity (as measured by EDS) while *cis*-eGenes tend towards lower than average EDS [3,4,20]. Focusing *trans*-QTL analyses on expression regulators and candidate MRs resulted in even higher EDS distributions, which may be advantageous for explaining the effects of GWAS signals. Therefore, the relative advantage of aQTLs over eQTLs in *trans*, combined with the reduced multiple testing

burden achieved through the biologically motivated restriction of *trans*-QTL analyses to putative phenotypic MRs that likely mediate the flow of information through the regulatory network results in a higher chance of detecting downstream molecular effects mediating the effects of GWAS signals. This allows progress in the explanation of GWAS signals that currently lack any plausible, mediating *cis* effects.

Finally, at the 1p36.1 BMI locus we observed a situation where the *EPHB2 cis*-aQTL was stronger than its eQTL and colocalized not only with the BMI GWAS locus, but also with *trans*-aQTLs for 35 BMI MRs (15 of which were significant at FDR < 0.05). There are several potential explanations for why *EPHB2* has a stronger *cis*-aQTL than eQTL, including allelic effects on transcript traits or transcription dynamics that have little impact on steady state transcript levels, as discussed above. However, in this case it may in part be due to EPHB2's identification as a putative BMI MR. Consequently, its expression and activity were highly correlated with that of many other putative BMI MRs. Considering this and the observed colocalizations between the *EPHB2 cis*-aQTL and many significant MR *trans*-aQTLs, there may be some feedback mechanism at play that amplifies the observed effect on EPHB2's activity through the *trans* effects on the activities of other BMI MRs. Functional studies are required to further explore this phenomenon. Regardless, to our knowledge, prior to this study there was no proposed explanation for the molecular effects that may mediate the BMI GWAS signal at chr1p36.1. The notion of *EPHB2* (which encodes the bi-directional ephrin receptor B2) as a MR of the BMI-associated adipose cell state is consistent with the literature that has linked *EPHB2* expression to lipid metabolism in prostate tumor cells and adipose ephrin signaling to obesity in mice [39,40].

In summary, by using only existing adipose eQTL datasets we were able to leverage the gene regulatory landscape inferred from the expression data to infer expression regulator activities, identify adipose phenotypic master regulators (MRs) matched or relevant to several cardiometabolic traits, and detect activity QTLs (aQTLs) that colocalized with cardiometabolic trait GWAS signals. Activity QTLs were often stronger than their eQTLs among MRs in *trans*, but occasionally *cis*-aQTLs were also informative. Altogether, our MR and aQTL approach enabled the generation of new hypotheses for molecular effects mediating GWAS signals for complex traits that could be followed up in future functional studies.

## Materials and methods

### Transcriptome, genotype, phenotype and GWAS datasets

Subcutaneous adipose RNA-seq, imputed genotype and phenotype data were obtained via controlled access from the TwinsUK resource for 766 female, twin subjects of European descent [16,17]. The TwinsUK RNA samples were sequenced and quantified as described previously [17]. In brief, reads were mapped to GENCODE version 10 and normalized by median number of well-mapped reads. We received the RNA-seq results as RPKM values, which we converted to TPM values and $\log_2$ transformed. Genes were excluded for low expression by the following scheme: < 5% samples with TPM greater than the minimum scaling factor across all samples. The scaling factor for each sample was calculated as the ratio of average TPM to average RPKM across all genes.

Genotype data for the TwinsUK samples were generated from various arrays (HumanHap300, HumanHap610Q, 1M-Duo and 1.2MDuo Illumina arrays) and imputation was performed against the 1000 Genomes phase 1 reference panel using IMPUTE2 as previously described [17,41,42].

BMI, waist and hip measurements (in cm), blood HDL, LDL and triglycerides (in mmol/L) and fasting insulin (in pmol/L) and glucose (in mmol/L) were provided for most of the

TwinsUK samples. Homeostatic model assessment of insulin resistance (HOMA-IR) was calculated from fasting insulin and glucose using the HOMA2 Calculator (https://www.dtu.ox.ac.uk/homacalculator/) and then natural log transformed [43]. All the above phenotypes were measured at time of biopsy except the waist and hip measurements that were taken 7–16 years earlier. In analyses using waist-hip ratio (WHR), we only used subjects whose BMI varied by less than 10% between the time of waist/hip measurements and adipose biopsy.

METSIM study gene expression and phenotype data were downloaded from the Gene Expression Omnibus (GEO) [19]. Expression array probe data was consolidated to gene level expression with the limma R package using the avereps() function [44].

We used genome-wide association studies (GWAS) for body-mass index (BMI), waist-hip ratio (BMI-adjusted WHR), type 2 diabetes (BMI-adjusted T2D), plasma high density lipids (HDL) and plasma triglycerides from the European population [45–48].

## Ancestry analysis

Initially, unrelated subjects were identified from the TwinsUK samples using KING (Kinship-based Inference for GWAS) software and these unrelated subjects were combined with the European references from the HGDP and the 1000 Genomes Project phase 3 [49–53]. The combined data were LD pruned and used for admixture analysis to calculate admixture proportions along with the European references using ADMIXTURE software [54]. Based on the cross-validation error estimate, K = 3 was used for the model. The excluded related subjects from the TwinsUK samples were subsequently projected to the previous ADMIXTURE result. Principal component (PC) analysis of the TwinsUK data was performed using TRACE software [55]. Briefly, the reference PCA was constructed from the European reference population from the HGDP and the 1000 Genomes Project phase 3, then the TwinsUK data was projected to the reference PCA.

## Gene regulatory network inference

The adipose gene co-expression network was inferred using the Algorithm for the Reconstruction of Accurate Cellular Networks with adaptive partitioning (ARACNe-AP; https://github.com/califano-lab/ARACNe-AP) [15]. This method uses mutual information (MI) to quantify the dependence between each pair of genes to construct co-expression networks. It consists of three steps: 1) Estimate the significance threshold of MI values from the given expression profiles, 2) Calculate bootstrap MI networks for randomly sampled subsets of given expression profiles, and 3) Build a consensus network from the bootstrap networks. We ran the ARACNe-AP network inference on 766 TwinsUK adipose expression profiles (in $\log_2$TPM) with 900 bootstraps and the default $P$-value threshold of $1 \times 10^{-8}$. The source for each edge (i.e. the regulator) was assigned according to the expression regulator list (6,153 genes listed in **S1 Table**) that included transcription factors, transcription co-factors and signal transduction factors. The final consensus network contained 4,221 regulators and 13,775 targets with a total of 730,059 directed edges (**S2 Table**).

## Gene expression regulator activity inference

Expression regulator activity inference was performed using Virtual Inference of Protein Activity by Enriched Regulon analysis (VIPER; http://bioconductor.org/packages/release/bioc/html/viper.html) [18]. First, the ARACNe adipose co-expression network was converted into an interactome by the aracne2regulon function within VIPER by the following calculations: 1) the mode of action (i.e. weighted direction of effect) between each regulator and each of its targets based on their Spearman correlations; 2) the likelihood of the connection (**S2**

Table). Activity scores were then calculated from Z-scaled $\log_2$TPM gene expression levels (per gene across samples) based on the adipose interactome using the viper function.

## Phenotypic master regulator (MR) analysis

Putative phenotypic MRs for subcutaneous adipose, defined as a set of regulators whose regulatory activity scores in adipose samples best predict a given phenotype, were identified by the machine learning tool random forest regression (using the randomForest R package) via the following steps: 1) test cross-validation error for random forest models with increasing numbers of regulators to determine a parsimonious number of regulators sufficient to minimize prediction error; 2) identify the most important regulators as measured by the percent increase in the mean square error (MSE) upon permutation of the regulator in all trees of the forest; 3) train the final random forest model with the determined number of top regulators by importance (S1 Fig). Training and test sets were formed as a 70:30 split, respectively, of the TwinsUK adipose regulator activity profiles. Phenotypic master regulators were identified for body mass index (BMI), waist-hip ratio (WHR), the natural log of the homeostatic model assessment of insulin resistance (HOMA-IR), plasma high-density lipids (HDL) and plasma triglycerides. Only adipose samples that were time-matched to the phenotype measurement were used in the MR analysis (388–699 samples), except for WHR, which was not measured at the time of the adipose tissue biopsy. To mitigate the noise that could potentially be introduced by changes in WHR, we restricted the WHR MR analysis to samples from subjects whose BMI varied by less than 10% from the time of WHR measurement to the time of biopsy.

## Expression and activity quantitative trait locus (QTL) analysis

For all QTL analyses, we used only one of each monozygotic twin pair and only subjects with >80% European admixture (699 subjects after these filters). Furthermore, the top 5 principal components from ancestry informative genotypes were included in all QTL analyses to adjust for any remaining population stratification. Age was the only other covariable included in the eQTL and aQTL linear models that were trained using Matrix eQTL [56]. Discovery eQTL and aQTL analyses were run separately for each GWAS and tested only SNPs with GWAS $P \leq 5 \times 10^{-8}$. *Cis* analyses included only expression regulator genes (for which activity scores were inferred) within 1Mb of the significant GWAS SNPs, while *trans* analyses included only the phenotypic MRs matched to the GWAS (i.e. BMI MRs for BMI GWAS, WHR MRs for BMI-adjusted WHR GWAS, HOMA-IR MRs for BMI-adjusted T2D GWAS, HDL MRs for HDL GWAS and triglycerides MRs for triglycerides GWAS). The following multiple testing procedures were applied for the *cis* and *trans* QTL analyses:

*Cis*-QTL analysis for a given phenotype (e.g., BMI):

1. Identify all SNPs (denoted as **A,** N SNPs in total) achieving genome-wide significance.

2. For the phenotype, identify all expression regulator genes that are within 1Mb of any SNPs in **A**. We assume there are *K* genes and denote the gene set as **B**.

3. For each gene (k) in **B**, identify all SNPs (n) in **A** that are within the *cis*-region of the given gene k. Suppose the *P*-values for the QTL test are $p_{k1}, p_{k2}, \ldots, p_{kn}$. Let $P_k = \min_{1 \leq i \leq n} p_{ki}$ be the minimum *P*-value of the n SNPs in the *cis* region for gene k.

4. We performed 100,000 simulations using the genotype of the n SNPs in the subjects from the 1000 Genome project to generate the empirical distribution of $P_k$, which was used to generate the gene-wise *P*-value (denoted as $q_k$) for the gene after accounting for multiple testing of the n SNPs in the *cis*-region.

5. We used the BH procedure to identify significant regulator genes controlling FDR ≤ 5%.

*Trans*-QTL analysis for a given phenotype (e.g. BMI):

1. Identify all MR genes ($L$) for the phenotype.

2. For a given MR gene $l$, suppose the *P*-values of the QTL analysis were $p_{l1}, \cdots, p_{lN}$. Let $P_l = \min_{i \in A} p_{li}$ be the minimum *P*-value in the QTL analysis for the MR gene.

3. We performed 100,000 simulations using the genotype of the N SNPs from the 699 European, unrelated TwinsUK subjects to generate the empirical distribution of $P_l$ and derived the gene-wise *P*-value (denoted as $q_k$) for the gene.

4. We used the BH procedure to identify significant MR genes controlling FDR ≤ 5%.

5. We identified the *trans*-QTL significance threshold as the nominal *P*-value corresponding to the FDR-adjusted gene-wise *P* = 0.05.

For the loci listed in **Table 1** plus the BMI locus at 17p13.2, eQTL and aQTL analyses were also rerun on the SNPs reported in the summary statistics of all five GWAS (regardless of GWAS *P*) that were within a 1Mb window centered on the top GWAS variant against all MRs and all *cis* genes (though aQTLs were again restricted to expression regulators). These focused, dense eQTL and aQTL results were used only for colocalization analyses rather than discovery, and therefore not corrected for multiple testing.

## Colocalization analysis

Colocalization analyses were performed with the HyPrColoc R package (https://github.com/jrs95/hyprcoloc) for the 25 distinct loci reported in **Table 1** plus the BMI locus at 17p13.2 using the dense eQTL and aQTL results (described above), the GWAS summary statistics and LD matrices for the loci calculated from the TwinsUK imputed genotype data for the 699 samples included in the QTL analyses [23]. Only those SNPs present in the QTL analyses and all GWAS were included in the colocalization analyses. Pairwise colocalization was tested between all pairs of significant GWAS signals, between all pairs of significant GWAS signals and tested *cis* and *trans* eQTLs and aQTLs and between select matched eQTLs and aQTLs. We used the default posterior probability (*PP*) threshold of 0.25 but distinguished between signal pairs that colocalize weakly ($0.25 < PP < 0.50$) and those that colocalize well ($PP > 0.50$).

## Conditional analysis

For conditional analyses of select GWAS signals we used the conditional and joint analysis (COJO) method as implemented in the GCTA tool, which only requires summary statistics [24,25]. The 1p36.1 BMI GWAS signal was conditioned on rs6692586, rs4654828 and rs12408468 individually and in all combinations. The chr7q32.2 BMI, BMI-adjusted T2D, HDL and triglycerides GWAS signals were conditioned on rs972283, rs287621 and rs738134 individually and in all combinations. These SNPs were chosen to represent distinct LD blocks within the loci among the 1000 Genomes European population [41].

## Generation of figures, plots and graphs

All scatter plots, box plots and violin plots were produced in R using plotting functions from both base R and the ggplot2 package. Random forest importance score plots were generated in R by the importance function of the randomForest package. Linkage disequilibrium (LD) plots showing pairwise $r^2$ between variants were generated by the LDmatrix tool within the

LDlink suite of web-based application (https://ldlink.nci.nih.gov/?tab=home), and were based on the 1000 Genomes EUR population [41,57]. Heatmaps were generated with the base R heatmap function from Pearson correlation matrices between the indicated expression or activity values that were also calculated in R. LocusZoom and LocusCompare plots were generated in R with the locuscomparer package [58]. All R scripts are available on GitHub (https://github.com/hoskinsjw/aQTL2021). Network graphs in **Fig 6** were generated in Cytoscape v3.8.1 with node border colors continuously mapped by Pearson correlations between BMI and regulator log$_2$TPMs or activity scores, node colors continuously mapped by eQTL or aQTL model betas, and edge colors continuously mapped by mode of action for regulator connections in the adipose interactome described in **S3 Table**.

## Supporting information

**S1 Fig. Master regulator analysis workflow.** First, test cross-validation error for random forest models with increasing numbers of regulators to determine a parsimonious number of regulators sufficient to minimize prediction error. Next, identify the most important regulators as measured by the percent increase in the mean square error (MSE) upon permutation of the regulator in all trees of the forest. Finally, train the final random forest model with the determined number of top regulators by importance and test the model in the test set and validation set.
(PDF)

**S2 Fig. Adipose BMI master regulator analysis. (A)** Cross-validated prediction performance of random forest regression models with the number of predictors sequentially reduced by five. Models were trained to predict BMI from regulator activities 12 times, each with a unique seed. The plot compares the number of predictors included in the model versus the mean cross-validation error and error bars indicate the standard deviation of the 12 analyses. **(B)** Rank order for the top 100 regulators by importance to BMI prediction by the final random forest model. The importance is measured as the percent increase in the mean squared error (MSE) upon permutation of the regulator across all trees of the random forest.
(PDF)

**S3 Fig. Adipose WHR master regulator analysis. (A)** Cross-validated prediction performance of random forest regression models with the number of predictors sequentially reduced by five. Models were trained to predict WHR from regulator activities 12 times, each with a unique seed. The plot compares the number of predictors included in the model versus the mean cross-validation error and error bars indicate the standard deviation of the 12 analyses. **(B)** Rank order for the top 100 regulators by importance to WHR prediction by the final random forest model. The importance is measured as the percent increase in the mean squared error (MSE) upon permutation of the regulator across all trees of the random forest.
(PDF)

**S4 Fig. Adipose HOMA-IR master regulator analysis. (A)** Cross-validated prediction performance of random forest regression models with the number of predictors sequentially reduced by five. Models were trained to predict ln(HOMA-IR) from regulator activities 12 times, each with a unique seed. The plot compares the number of predictors included in the model versus the mean cross-validation error and error bars indicate the standard deviation of the 12 analyses. **(B)** Rank order for the top 100 regulators by importance to ln(HOMA-IR) prediction by the final random forest model. The importance is measured as the percent increase in the mean squared error (MSE) upon permutation of the regulator across all trees of the random forest.
(PDF)

**S5 Fig. Adipose HDL master regulator analysis.** **(A)** Cross-validated prediction performance of random forest regression models with the number of predictors sequentially reduced by five. Models were trained to predict HDL from regulator activities 12 times, each with a unique seed. The plot compares the number of predictors included in the model versus the mean cross-validation error and error bars indicate the standard deviation of the 12 analyses. **(B)** Rank order for the top 100 regulators by importance to HDL prediction by the final random forest model. The importance is measured as the percent increase in the mean squared error (MSE) upon permutation of the regulator across all trees of the random forest.
(PDF)

**S6 Fig. Adipose Triglycerides master regulator analysis.** **(A)** Cross-validated prediction performance of random forest regression models with the number of predictors sequentially reduced by five. Models were trained to predict triglycerides from regulator activities 12 times, each with a unique seed. The plot compares the number of predictors included in the model versus the mean crossvalidation error and error bars indicate the standard deviation of the 12 analyses. **(B)** Rank order for the top 100 regulators by importance to triglycerides prediction by the final random forest model. The importance is measured as the percent increase in the mean squared error (MSE) upon permutation of the regulator across all trees of the random forest.
(PDF)

**S7 Fig. Comparisons between phenotypic master regulators in adipose.** **(A)** Clustered heatmap of Pearson correlations between the expression profiles for putative phenotypic master regulators in the TwinsUK adipose samples. The black bars to the right and bottom of the heatmap indicate for which phenotype each regulator was identified as a candidate master regulator. **(B)** Same as **A**, but between the activity profiles for putative phenotypic master regulators in the TwinsUK adipose samples. **(C)** Same as **A**, but between the expression profiles (for regulators indicated by row)and activity profiles (for regulators indicated by column) of putative phenotypic master regulators in the TwinsUK adipose samples. Note that the correlation coefficients for the heatmap in S7C Fig are not symmetrical about the diagonal since correlation between the expression profile of a row regulator and the activity profile of a column regulator is not the same as the converse, except along the diagonal (where the row and column regulators are identical). Consequently, though the rows and columns are in the same order, the clustering is based solely on columns (i.e. the vectors of correlation coefficients for the column regulators' activity profiles versus the expression profiles for the row regulators). **(D)** Density plot of Pearson correlation between matched expression and activity for master regulators. **(E)** Density plot of Pearson correlation between unmatched expression and activity for master regulators. **(F)** Venn diagram demonstrating the number of master regulators that are unique or in common among the phenotypes analyzed.
(PDF)

**S8 Fig. Pairwise colocalization of GWAS signals at chr7q32.2.** **(A)** LocusCompare plot of chr7q32.2 locus variants' $-\log_{10}(P)$ from BMI GWAS versus T2D (BMI-adjusted) GWAS. Points are colored according to the variant's $r^2$ with the reference variant, which is marked by the purple diamond. Posterior probability (*PP*) of a single, shared functional variant was calculated with HyPrColoc. **(B)** Same as **A**, but for BMI GWAS versus HDL GWAS. **(C)** Same as **A**, but for BMI GWAS versus Triglycerides GWAS. **(D)** Same as **A**, but for T2D (BMI-adjusted) GWAS versus HDL GWAS. **(E)** Same as **A**, but for T2D (BMI-adjusted) GWAS versus

Triglycerides GWAS. **(F)** Same as **A**, but for HDL GWAS versus Triglycerides GWAS.
(PDF)

**S9 Fig. Pairwise colocalization of GWAS signals with *LINC-PINT* and *AC016831.7 cis*-eQTLs at chr7q32.2. (A)** LocusCompare plot of chr7q32.2 locus variants' -log$_{10}$($P$) from BMI GWAS versus LINC-PINT *cis*-eQTL. Points are colored according to the variant's r$^2$ with the reference variant, which is marked by the purple diamond. Posterior probability (*PP*) of a single, shared functional variant was calculated with HyPrColoc. **(B)** Same as **A**, but for T2D (BMIadjusted) GWAS versus *LINC-PINT cis*-eQTL. **(C)** Same as **A**, but for HDL GWAS versus *LINC-PINT cis*-eQTL. **(D)** Same as **A**, but for Triglycerides GWAS versus *LINC-PINT cis*-eQTL. **(E)** Same as **A**, but for *LINC-PINT cis*-eQTL versus *LINC-PINT cis*-aQTL. **(F)** Same as **A**, but for BMI GWAS versus *AC016831.7 cis*-eQTL. **(G)** Same as **A**, but for T2D (BMI-adjusted) GWAS versus *AC016831.7 cis*-eQTL. **(H)** Same as **A**, but for HDL GWAS versus *AC016831.7 cis*-eQTL. **(I)** Same as **A**, but for Triglycerides GWAS versus *AC016831.7 cis*-eQTL.
(PDF)

**S10 Fig. BMI GWAS signal at chr12p13.1. (A)** Plot of genomic position versus BMI GWAS -log$_{10}$($P$) for variants at the chr12p13.1 locus. **(B)** LD plot between top nine BMI GWAS variants at the chr12p13.1 locus shaded by pairwise r$^2$.
(PDF)

**S1 Table. List of expression regulators.**
(XLSX)

**S2 Table. TwinsUK subcutaneous adipose ARACNe co-expression network.**
(ZIP)

**S3 Table. TwinsUK subcutaneous adipose interactome derived from the ARACNe co-expression network.**
(ZIP)

**S4 Table. Phenotypic master regulator (MR) list with pleiotropy count.**
(XLSX)

**S5 Table. cis-eQTL and aQTL results between BMI GWAS SNPs and genes with at least one significant QTL at FDR < 0.05.**
(XLSX)

**S6 Table. cis-eQTL and aQTL results between BMI-adjusted WHR GWAS SNPs and genes with at least one significant QTL at FDR < 0.05.**
(XLSX)

**S7 Table. cis-eQTL and aQTL results between BMI-adjusted T2D GWAS SNPs and genes with at least one significant QTL at FDR < 0.05.**
(XLSX)

**S8 Table. cis-eQTL and aQTL results between HDL GWAS SNPs and genes with at least one significant QTL at FDR < 0.05.**
(XLSX)

**S9 Table. cis-eQTL and aQTL results between Triglycerides GWAS SNPs and genes with at least one significant QTL at FDR < 0.05.**
(XLSX)

**S10 Table. cis-aRegulators significant at FDR < 0.05.**
(XLSX)

**S11 Table. trans-eQTL and aQTL results between BMI GWAS SNPs and BMI master regulators (MRs) with at least one significant QTL at the locus at FDR < 0.05.**
(XLSX)

**S12 Table. trans-eQTL and aQTL results between BMI-adjusted WHR GWAS SNPs and WHR master regulators (MRs) with at least one significant QTL at the locus at FDR < 0.05.**
(XLSX)

**S13 Table. trans-eQTL and aQTL results between BMI-adjusted T2D GWAS SNPs and HOMA-IR master regulators (MRs) with at least one significant QTL at the locus at FDR < 0.05.**
(XLSX)

**S14 Table. trans-eQTL and aQTL results between HDL GWAS SNPs and HDL master regulators (MRs) with at least one significant QTL at the locus at FDR < 0.05.**
(XLSX)

**S15 Table. trans-eQTL and aQTL results between Triglycerides GWAS SNPs and Triglycerides master regulators (MRs) with at least one significant QTL at the locus at FDR < 0.05**
(XLSX)

**S16 Table. Summary statistics for EDS distributions of gene sets from Fig 2.**
(XLSX)

**S17 Table. HyPrColoc pairwise co-localization results from dense cis- and trans-eQTL and aQTL analyses of loci in Table 1 (plus BMI locus 17p13.2).**
(XLSX)

**S18 Table. COJO conditional analyses of the 7q32.2 BMI GWAS signal.**
(XLSX)

**S19 Table. COJO conditional analyses of the 7q32.2 T2D (BMI-adjusted) GWAS signal.**
(XLSX)

**S20 Table. COJO conditional analyses of the 7q32.2 HDL GWAS signal.**
(XLSX)

**S21 Table. COJO conditional analyses of the 7q32.2 Triglycerides GWAS signal.**
(XLSX)

**S22 Table. COJO conditional analyses of the 1p36.1 BMI GWAS signal.**
(XLSX)

## Acknowledgments

This work used the high-performance computational capabilities of the Biowulf Linux cluster at the NIH, Bethesda, MD (https://hpc.nih.gov). We thank the TwinsUK cohort for providing the RNA-seq, genotype and phenotype data for the adipose samples used in this study, especially Dr. Kerrin Small for her insights on the data and sound advice. TwinsUK is funded by the Wellcome Trust, Medical Research Council, European Union, Chronic Disease Research

Foundation (CDRF), Zoe Global Ltd and the National Institute for Health Research (NIHR)-funded BioResource, Clinical Research Facility and Biomedical Research Centre based at Guy's and St Thomas' NHS Foundation Trust in partnership with King's College London.

## Author Contributions

**Conceptualization:** Jason W. Hoskins.

**Formal analysis:** Jason W. Hoskins, Charles C. Chung, Jianxin Shi.

**Funding acquisition:** Laufey T. Amundadottir.

**Methodology:** Jason W. Hoskins, Jianxin Shi.

**Project administration:** Jason W. Hoskins.

**Software:** Jason W. Hoskins.

**Supervision:** Laufey T. Amundadottir.

**Visualization:** Jason W. Hoskins.

**Writing – original draft:** Jason W. Hoskins.

**Writing – review & editing:** Jason W. Hoskins, Aidan O'Brien, Jun Zhong, Katelyn Connelly, Irene Collins, Jianxin Shi, Laufey T. Amundadottir.

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
