## [Decision Letter · Decision Letter 0]

1 Jul 2021

Dear Dr. Hoskins,

Thank you very much for submitting your manuscript "Inferred expression regulator activities suggest genes mediating cardiometabolic genetic signals" for consideration at PLOS Computational Biology.

As with all papers reviewed by the journal, your manuscript was reviewed by members of the editorial board and by several independent reviewers. In light of the reviews (below this email), we would like to invite the resubmission of a significantly-revised version that takes into account the reviewers' comments.

Please fully address the questions and comments raised by the reviewers. In addition, please make sure that all data and computational code underlying the findings in the manuscript are fully available.

We cannot make any decision about publication until we have seen the revised manuscript and your response to the reviewers' comments. Your revised manuscript is also likely to be sent to reviewers for further evaluation.

Sincerely,

Xiaoqiang Sun

Guest Editor

PLOS Computational Biology

Ilya Ioshikhes

Deputy Editor

PLOS Computational Biology

Please fully address the questions and comments raised by the reviewers. In addition, please make sure that all data and computational code underlying the findings in the manuscript are fully available.

Reviewer's Responses to Questions

**Comments to the Authors:**

Reviewer #1: Hoskins et al. leveraged an adipose-specific gene regulatory network to infer expression regulator activities and phenotypic master regulators (MRs), which were used to detect activity QTLs (aQTLs) at cardiometabolic trait GWAS loci and reported several GWAS loci colocalized with MR trans- eQTLs/aQTLs in the absence of colocalized cis-QTLs. The transformation of gene expression into activity by integrating information including gene-gene interactions may be informative for genetic researchers and provide insights for downstream analysis of GWAS loci. But some of the results in this study seem to be biased, there are some major comments as below.

1. Regulator activities were inferred from the expression data and the adipose co-expression network which was constructed using TwinsUK RNA-seq data of female samples. As far as I am concerned, estrogen has a protective effect on cardiometabolic phenotypes, which would introduce more confounding factors in this study if the regulator activities of regulatory genes were inferred solely from female data. In the “Inference of gene regulator activities from transcriptomic data” part, would it be better if the TwinsUK dataset and METSIM dataset were merged together to infer regulator activity with adjustment for sex or simply use METSIM expression array dataset which was all male samples as training set and TwinsUK as test set?

2. In trans-eQTL and aQTL analysis of GWAS signals, the multiple testing burden of the analysis was reduced by restricting the target genes to MRs of matching or relevant phenotypes. It is reasonable for the analysis for trans-aQTL while not appropriate for trans-eQTL because the MRs were inferred from the activities but not the expression of regulatory genes. Trans-aQTLs tend to outperform their respective eQTLs because of the biased MRs selection.

Reviewer #2: Hoskins et.al describe the application of aQTL analysis to identify alleles associated with cardiometabolic traits in fat tissue, including BMI, fat distribution, diabetes risk and blood cholesterol levels. While the global approach is not original, limiting the search to the genes/proteins most likely causally associated to the phenotype of interest dramatically boosts the statistical power of the GWAS/aQTL analysis and enrich the results in driver vs. passenger genes, allowing the discovery of potentially causal genetic-to-phenotypic associations that has not been previously described. This work constitutes an important contribution to the GWAS field. aQTL has the potential to transform GWAS analysis by providing regulatory-based mechanistic links between genomic and phenotypic traits and I expect to see many more reports using aQTL analysis.

Mayor points

1. Page #5, line 97: The authors state "Toward this end, we have developed a new method called activity QTL (aQTL) analysis that leverages tissue-specific gene expression regulatory networks to identify genetic effects on expression regulatory activities". However, the method they refer to is not new and have been previously described. See for example Chen et.al (2014). Identification of Causal Genetic Drivers of Human Disease through Systems-Level Analysis of Regulatory Networks. Cell, 159(2), 402–414. https://doi.org/10.1016/j.cell.2014.09.021. In fact, one of the steps of the DIGGIT algorithm, described in the reference above, is precisely aQTL analysis. This approach has been also used in other publications, such as Paull et.al. Cell 184:334 (2021), Bisikirska et.al. Cancer Res. 76:664 (2016), and Thorsson et.al. Immunity 48:812 (2018), among others. This previous work should be mentioned in the introduction and differences in the implementation with the author's method should be discussed.

2. Page #8, line164: "Notably, while there is a high degree of correlation between these regulator's expression levels and respective activities, is is far from perfect, which highlights the important difference between these two metrics (Fig. S7C).". The authors need to be more specific. What does "high degree of correlation" means? What does "it is far from perfect" means? How the authors define "perfect" and "far"?

In addition, Figure S7C is impossible to interpret given the rows and columns have been clustered using different similarity inferences. This can be better displayed by clustering the genes based only on expression, only on inferred protein activity, or on an integration of both, but keeping the order of rows and columns the same, as to see the paired (gene-matched) correlation between expression and activity inferences in the diagonal of the heatmap. A different, maybe more effective way to show this data would be by plotting the distributions of the correlation coefficient for the gene-matched pairs and include in the same figure, as reference, the density distribution for the correlation coefficient for all the un-matched pairs. This should show that while there is strong correlation between expression and activity for some proteins, there will also be exceptions, specially for proteins whose activity is strongly post-translationally regulated.

3. Page #8, line 172: "...variants with genome-wide significant (i.e. P < 5E-8) associations...". Why this particular p-value threshold? Why not 1E-10 or 1E-5 for example? Multiple hypothesis correction?

4. Please, consider referring to the MR proteins as "candidate MRs", since while their activity is statistically associated with the phenotype, they have not been experimentally validated as bona fide, causal drivers of the phenotype.

Minor points

1. Page #6, line 110: "Network edges in ARACNe co-expression networks are directed from regulator to target". This statement is not accurate. Since ARACNe edges are defined based on Mutual Information (MI), the edges are not directed. When the gene pair is constituted by a transcriptional regulator (TR) and non-TR protein, we can assume the direction is TR -> non-TR. However, when both proteins as TR or both are non-TR, the direction (i.e. which is the regulator and which is the target) is not clear.

2. Page #7, line 143: "To further validate these phenotypic MRs for adipose, we used the METSIM expression..." This approach does not validate the candidate MRs as regulators of the phenotype, but test their association, and the RandomForest-model predictions, on an independent dataset. So I would suggest rephrasing to "To further test the candidate MRs on an independent dataset, we used the METSIM..."

3. Page #8, line 161: The authors mention that the lack of clustering among MRs of the same phenotype suggest pleiotropy among MRs of these well correlated cardiometabolic phenotypes. However I do not think this is due too pleiotropy but simply because the cardiometabolic phenotypic traits represent only a minor component of the phenotypic (protein activity) variance of the dataset, and an unsupervised analysis as the one presented in Fig. S7B would capture only the major sources of phenotypic variance. The authors could corroborate this by performing a principal components analysis and checking the proportion of variance captures by the components showing significant association with the cardiometabolic phenotypes of their interest.

4. Page #19, line 404: previously

5. Consider adding to the discussion that by limiting the search for association to the candidate MRs of a particular phenotypic trait, aQTL analysis not only boosts the statistical power to find significant associations, but also does it for genes/proteins likely having a role as mechanistic regulators of the phenotype.

Reviewer #3: Review: Inferred expression regulator activities suggest genes mediating cardiometabolic genetic signals

Overview:

Hoskins et al. introduce a new method called activity QTL (aQTL) analysis to identify loci that have genetic effects on expression regulatory activities. This method is inspired by the issue that cis-eQTL only explains portion of GWAS mediating signals of complex traits and integrating trans-eQTL signals is underpowered due to large multiple testing burdens. As a result, authors hypothesize that GWAS functional variants have detectable impacts on the regulatory activities as reflected by X. Authors construct a regulatory network using transcriptomic data and expression regulators (TF, T co-F, etc.) and then inferred regulator activities scores. Using this network, the authors identify “master regulators” (MRs) for downstream trans-QTL analyses by prioritizing activities that predict cardiometabolic phenotypes. Then, authors performed cis-e/aQTL analyses on all expression regulator genes within 1MB of the significant GWAS SNPs and trans-e/aQTL analyses on MRs matched to the GWAS.

I find this area of research to be timely and important and the proposed approach interesting. The out-of-sample replication of random forests in METSIM data was particularly compelling. My comments are related mostly to providing additional overview/summary information of the main findings to help provide context for the interesting cases before discussing individual findings.

Major Comments:

1. In the abstract and introduction, it is unclear to me what quantity activity refers to specifically and how it is different from measured expression levels. Given, the central importance of this concept I think spending more space to providing appropriate context and definitions would be very helpful.

2. The cis e/aQTL analysis in main text lacks summary information that would provide context to their findings. Namely, the total number of variant/gene pairs tested and the number of identified cis e/aQTLs identified at their given FDR level. (While table 1 alleviates this somewhat for trans e/aQTLs, a simple statement in the main text providing the total number of variant/gene pairs tested and the number of trans e/aQTLs identified would be helpful).

3. The authors note “two notable exceptions” where cis-aQTLs were stronger than correspond cis-eQTL effect sizes, however, it is not clear if two differences is expected or not, given the total number of tests performed (see above comment).

4. In comparing the proportions of identified QTLs with, the authors report a difference of 20% (cis) vs 72% (trans) of aQTLs having smaller p-values in compared to their eQTL counterparts. Can the authors attach a p-value to this statement using a proportion test or something similar? It isn’t clear what to expect based on the total number of identified cis vs trans QTLs.

5. In the colocalization analysis, it would be helpful to state directly (or in a table) the total number of risk regions examined, the total number of e/aQTLs that colocalized at some threshold (stratified by in cis vs in trans).

6. Recent work (Wang et al AJHG 2020; 10.1016/j.ajhg.2020.01.012) has demonstrated that GWAS signals are enriched for regions with increased enhancer complexity. It would be interesting to contrast enrichment of cis/trans aQTLs across respective EDS scores (see ref) compared with cis/trans eQTL. Namely, do colocalizing aQTLs do a better job characterizing risk regions with complex enhancer activity compared with eQTLs?

Minor Comments:

1. Line 26 of the abstract: “outperformed” is slightly ambiguous. If it reflects identifying “explanatory genes”, I recommend being more explicit.

2. While not critical, it would be interesting to perform an set enrichment or ontology analysis on the identified MR genes to provide context for putative function/pathways.

Reviewer #4: Hoskins et.al., proposed an “activity QTL(aQTL) analysis” and hypothesized that genetic variants regulating disease-associated master regulators(MR) could underlie the GWAS signals for metabolic traits. The authors constructed gene co-expression network in forms of regulator-> targets connections at subcutaneous adipose, a tissue highly relevant to metabolic traits. They then inferred activities of regulators from the network. They next applied random forest regression to identify top regulators(MR) for each metabolic trait. With those information they performed analyses to identify cis/trans eQTLs/aQTLs that potentially regulate the expression or the activity of MRs in subcutaneous adipose and to test whether and how those QTLs colocalize with GWAS hits of metabolic traits. They found whilst in cis, regulatory variants that colocalize with GWAS hits seem to have a stronger effect for expression of MRs than for activity of MRs, in trans those variants tend to have a stronger effect for activity than for expression of MRs. They also provided examples where MR trans-QTLs(eQTLs or aQTLs) provided intriguing clues of the functional relevance of GWAS loci when cis-QTLs provided limited information. The authors concluded that these analyses can be broadly applied given the availability of data(expression and GWAS data) for many complex traits, and that expression/activity-type analyses of MRs add additionally valuable findings to classical global eQTL analyses.

Overall, I think this study proposed advanced and interesting concepts (aQTLs, MRs) and provided promising application (colocalization with GWAS signals). The analyses were carefully designed and the results are appropriately discussed. I have some questions which would like the authors to consider:

1. The author claimed that overall trans-aQTLs tend to be more significant/stronger than their respectively eQTLs. Was this conclusion drawn from raw p values from QTL analysis, or has the effect size been compared(if both activity and expression have been normalized onto comparable scale)? According to the paper, in its nature the activity of a MR reflected expression input from its target gene members within the regulon. Would it be possible that a strong tran-aQTL signal of a MR simply tracked a strong cis-eQTL signal of one of the target genes in the regulon of that MR?

2. A relevant and important question is that which variants, those interrupting/regulating MR or those interrupting/regulating MRs’ target genes in the regulon contribute more to the genetic susceptibility of complex traits? There has been debate in the field about the omnigenetic model(or the core-gene model) https://www.ncbi.nlm.nih.gov/pmc/articles/PMC5536862/ and the polygenicity model https://pubmed.ncbi.nlm.nih.gov/29906445/ for genetic architecture of complex traits. What the authors explored here are essentially in line with the ‘omnigenetic model’ by showing variants regulating core-genes(master regulators) with potentially larger effect can underlie some GWAS signals. What the author haven’t explored was that whether variants in target genes of the MRs also contributed to GWAS signals (polygenicity model), and how does that compare to the contribution from MR QTLs?. In figure s7, the author observed extensive correlation between MRs, both in terms of activity and expression. This suggested relatively limited independent pathways/room to be interrupted at core-genes(MRs) level and implicate peripheral genes(target genes of MRs) can potentially be better candidates for GWAS signals.

My other minor comments/questions are:

1. L150: whether the gender differences are responsible for the performance differences can be directly tested by only including females in test dataset. HDL and Triglycerides are potentially more fluctuated than other traits though.

2. In some places the authors used ‘overlapped’ in other places they used ‘colocalized’. Are they interchangeable? Please be consistent.

3. The snp id is different in L214, legends of Figure 2 and the figure 2 itself. Why among the two independent SNPs, only one was considered?

**Have the authors made all data and (if applicable) computational code underlying the findings in their manuscript fully available?**

Reviewer #1: None

Reviewer #2: Yes

Reviewer #3: Yes

Reviewer #4: None

PLOS authors have the option to publish the peer review history of their article (what does this mean?). If published, this will include your full peer review and any attached files.

Reviewer #1: No

Reviewer #2: **Yes: **Mariano J. Alvarez

Reviewer #3: No

Reviewer #4: No
---

## [Decision Letter · Decision Letter 1]

15 Oct 2021

Dear Dr. Hoskins,

We are pleased to inform you that your manuscript 'Inferred expression regulator activities suggest genes mediating cardiometabolic genetic signals' has been provisionally accepted for publication in PLOS Computational Biology.

Best regards,

Xiaoqiang Sun

Guest Editor

PLOS Computational Biology

Ilya Ioshikhes

Deputy Editor

PLOS Computational Biology

Reviewer's Responses to Questions

**Comments to the Authors:**

Reviewer #1: The concerns i had have been well addressed by the authors. There are no further comments.

Reviewer #2: The authors have addressed all my comments.

Reviewer #3: The authors have addressed my previous comments. I have no new comments at this time.

Reviewer #4: The author has addressed my questions.

**Have the authors made all data and (if applicable) computational code underlying the findings in their manuscript fully available?**

Reviewer #1: None

Reviewer #2: Yes

Reviewer #3: Yes

Reviewer #4: None

PLOS authors have the option to publish the peer review history of their article (what does this mean?). If published, this will include your full peer review and any attached files.

Reviewer #1: No

Reviewer #2: **Yes: **Mariano J. Alvarez

Reviewer #3: No

Reviewer #4: No

---

## [Editor Report · Acceptance letter]

12 Nov 2021

PCOMPBIOL-D-21-00772R1 

Inferred expression regulator activities suggest genes mediating cardiometabolic genetic signals

Dear Dr Hoskins,

I am pleased to inform you that your manuscript has been formally accepted for publication in PLOS Computational Biology. Your manuscript is now with our production department and you will be notified of the publication date in due course.

With kind regards,

Katalin Szabo
